# Guided Adversarial Attack for Evaluating and Enhancing Adversarial Defenses

**Gaurang Sriramanan**\*, **Sravanti Addepalli**\*, **Arya Baburaj**, **R.Venkatesh Babu**
Video Analytics Lab, Department of Computational and Data Sciences
Indian Institute of Science, Bangalore, India

## Abstract

Advances in the development of adversarial attacks have been fundamental to the progress of adversarial defense research. Efficient and effective attacks are crucial for reliable evaluation of defenses, and also for developing robust models. Adversarial attacks are often generated by maximizing standard losses such as the cross-entropy loss or maximum-margin loss within a constraint set using Projected Gradient Descent (PGD). In this work, we introduce a relaxation term to the standard loss, that finds more suitable gradient-directions, increases attack efficacy and leads to more efficient adversarial training. We propose *Guided Adversarial Margin Attack* (GAMA), which utilizes function mapping of the clean image to guide the generation of adversaries, thereby resulting in stronger attacks. We evaluate our attack against multiple defenses and show improved performance when compared to existing attacks. Further, we propose *Guided Adversarial Training* (GAT), which achieves state-of-the-art performance amongst single-step defenses by utilizing the proposed relaxation term for both attack generation and training.

## 1 Introduction

The remarkable success of Deep Learning algorithms has led to a surge in their adoption in a multitude of applications which influence our lives in numerous ways. This makes it imperative to understand their failure modes and develop reliable risk mitigation strategies. One of the biggest known threats to systems that deploy Deep Networks is their vulnerability to crafted imperceptible noise known as adversarial attacks, as demonstrated by Szegedy *et al.*[30] in 2014. This finding has spurred immense interest towards identifying methods to improve the robustness of deep neural networks against adversarial attacks. While initial attempts of improving robustness against adversarial attacks used just single-step adversaries for training [14], they were later shown to be ineffective against strong multi-step attacks by Kurakin *et al.*[22]. Some of the defenses introduced randomised or non-differentiable components, either in the pre-processing stage or in the network architecture, so as to minimise the effectiveness of generated gradients. However, many such defenses [4, 35, 29, 16] were later broken by Athalye *et al.*[3] using smooth approximations of the function during the backward pass or by computing reliable gradients using expectation over the randomized components. This game of building defenses against existing attacks, and developing attacks against the proposed defenses has been crucial for the progress in this field. Lately, the community has also recognized that the true testimony of a developed defense is to evaluate it against adaptive attacks which are constructed specifically to compromise the defense at hand [6].

Multi-step adversarial training is one of the best known methods of achieving robustness to adversarial attacks today [24, 37]. This training regime attempts to solve the minimax optimization problem of firstly generating strong adversarial samples by maximizing a loss, and subsequently training the

---

Correspondence to: Gaurang Sriramanan <gaurangs@iisc.ac.in>, Sravanti Addepalli <sravantia@iisc.ac.in>

model to minimize loss on these adversarial samples. The effectiveness of the defense thus developed depends on the strength of the attack used for training. Therefore, development of stronger attacks is important for both evaluating existing defenses, and also for constructing adversarial samples during adversarial training. Indeed, the study of building robust adversarial defenses and strong adversarial attacks are closely coupled with each other today.

Adversarial attacks are constructed by maximizing standard objectives such as cross-entropy loss or maximum-margin loss within a constraint set, as defined by the threat model. Due to the non-convex nature of the loss function, maximization of such a loss may not effectively find the path towards the class whose decision boundary is closest to the data point. In this work, we aid the optimization process by utilizing the knowledge embedded in probability values corresponding to non-maximal classes to guide the generation of adversaries. Motivated by graduated optimization methods, we improve the optimization process by introducing an $\ell_2$ relaxation term initially, and reducing the weight of this term gradually over the course of optimization, thereby making it equivalent to the primary objective towards the end. We demonstrate state-of-the-art results on multiple defenses and datasets using the proposed attack. We further analyse the impact of utilizing the proposed method to generate strong attacks for adversarial training. While use of the proposed attack for multi-step training shows only marginal improvement, we observe significant gains by using the proposed attack for single-step adversarial training. Single-step methods rely heavily on the initial gradient direction, and hence the proposed attack shows significant improvement over existing methods.

Our contributions in this work can be summarized as follows:

- We propose *Guided Adversarial Margin Attack* (GAMA), which achieves state-of-the-art performance across multiple defenses for a single attack and across multiple random restarts.

- We introduce a multi-targeted variant GAMA-MT, which achieves improved performance compared to methods that utilize multiple targeted attacks to improve attack strength [15].

- We demonstrate that Projected Gradient Descent based optimization (GAMA-PGD) leads to stronger attacks when a large number of steps (100) can be used, thereby making it suitable for defense evaluation; whereas, Frank-Wolfe based optimization (GAMA-FW) leads to stronger attacks when the number of steps used for attack are severely restricted (10), thereby making it useful for adversary generation during multi-step adversarial training.

- We propose *Guided Adversarial Training* (GAT), which achieves state-of-the-art results amongst existing single-step adversarial defenses. We demonstrate that the proposed defense can scale to large network sizes and to large datasets such as ImageNet-100.

Our code and pre-trained models are available here: https://github.com/val-iisc/GAMA-GAT.

## 2 Preliminaries

**Notation:** In this paper, we consider adversarial attacks in the setting of image classification using deep neural networks. We denote a sample image as $x \in \mathcal{X}$, and its corresponding label as $y \in \{1, \ldots, N\}$, where $\mathcal{X}$ indicates the sample space and $N$ denotes the number of classes. Let $f_\theta : \mathcal{X} \to [0, 1]^N$ represent the deep neural network with parameters $\theta$, that maps an input image $x$ to its softmax output $f_\theta(x) = \left( f_\theta^1(x), \ldots, f_\theta^N(x) \right) \in [0, 1]^N$. Further, let $C_\theta(x)$ represent the argmax over the softmax output. Thus, the network is said to successfully classify an image when $C_\theta(x) = y$. The cross-entropy loss for a data sample, $(x_i, y_i)$ is denoted by $\ell_{CE}(f_\theta(x_i), y_i)$. We denote an adversarially modified counterpart of a clean image $x$ as $\widetilde{x}$.

**Adversarial Threat Model:** The goal of an adversary is to alter the clean input image $x$ such that the attacked image $\widetilde{x}$ is perceptually similar to $x$, but causes the network to misclassify. Diverse operational frameworks have been developed to quantify perceptual similarity, and adversarial attacks corresponding to these constraints have been studied extensively. We primarily consider the standard setting of worst-case adversarial attacks, subject to $\ell_p$-norm constraints. More precisely, we consider adversarial threats bound in $\ell_\infty$ norm: $\widetilde{x} \in \{x' : \|x' - x\|_\infty \leq \varepsilon\}$.

While evaluating the proposed defense, we consider that the adversary has full access to the model architecture and parameters, since we consider the setting of worst-case robustness. Further, we assume that the adversary is cognizant of the defense techniques utilised during training or evaluation.

# 3 Related Works

## 3.1 Adversarial Attacks

A panoply of methods have been developed to craft adversarial perturbations under different sets of constraints. One of the earliest attacks specific to $\ell_\infty$ constrained adversaries was the Fast Gradient Sign Method (FGSM), introduced by Goodfellow *et al.*[14]. In this method, adversaries are generated using a single-step first-order approximation of the cross-entropy loss by performing simple gradient ascent. Kurakin *et al.*[21] introduced a significantly stronger, multi-step variant of this attack called Iterative FGSM (I-FGSM), where gradient ascent is iteratively performed with a small step-size, followed by re-projection to the constraint set. Madry *et al.*[24] developed a variant of this attack, which involves the addition of initial random noise to the clean image, and is commonly referred to as Projected Gradient Descent (PGD) attack.

Carlini and Wagner [5] explored the use of different surrogate loss functions and optimization methods to craft adversarial samples with high fidelity and small distortion with respect to the original image. The authors introduce the use of maximum margin loss for generation of stronger attacks, as opposed to the commonly used cross-entropy loss. Our proposed attack introduces a relaxation term in addition to the maximum margin loss in order to find more reliable gradient directions.

The Fast Adaptive Boundary (FAB) attack, introduced by Croce and Hein [9] produces minimally distorted adversarial perturbations with respect to different norm constraints, using a linearisation of the network followed by gradient steps which have a bias towards the original sample. While the FAB attack is often stronger than the standard PGD attack, it is computationally more intensive for the same number of iterations. Gowal *et al.*[15] introduced the Multi-Targeted attack, which cycles over all target classes, maximising the difference of logits corresponding to the true class and the target class. While this attack finds significantly stronger adversaries compared to PGD attack, it relies on cycling over multiple target classes, and hence requires a large computational budget to be effective. More recently, Croce and Hein [10] proposed AutoPGD, which is an automatised variant of the PGD attack, that uses a step-learning rate schedule adaptively based on the past progression of the optimization. They further introduce a new loss function, the Difference of Logits Ratio (DLR), which is a scale invariant version of the maximum margin loss on logits, and outperforms the $\ell_\infty$ based Carlini and Wagner (C&W) attack [5]. Additionally, they proposed AutoAttack, an ensemble of AutoPGD with the cross-entropy loss and the DLR loss, the FAB attack and Square attack [2], a score-based black-box attack which performs zeroth-order optimization.

## 3.2 Defenses Against Adversarial Attacks

With the exception of a few defenses [8, 1], most methods used to produce robust networks include some form of adversarial training, wherein training data samples are augmented with adversarial samples during training. Early works proposed training on FGSM [14], or Randomised FGSM (R-FGSM) [31] adversaries to produce robust networks. However, these models were still overwhelmingly susceptible to multi-step attacks [22] due to the Gradient Masking effect [25]. Madry *et al.*[24] proposed a min-max formulation for training adversarially robust models using empirical risk minimisation. It was identified that strong, multi-step adversaries such as Projected Gradient Descent (PGD), were required to sufficiently approximate the inner maximization step, so that the subsequent adversarial training yields robust models. Following this, Zhang *et al.*[37] presented a tight upper bound on the gap between natural and robust error, in order to quantify the trade-off between accuracy and robustness. Using the theory of classification calibrated losses, they develop TRADES, a multi-step gradient-based technique. However, methods such as TRADES and PGD-Training are computationally intensive, as they inherently depend upon the generation of strong adversaries through iterative attacks.

Consequently, efforts were made to develop techniques that accelerated adversarial training. Shafahi *et al.*[28] proposed a variant of PGD-training, known as Adversarial Training for Free (ATF), where the gradients accumulated in each step are used to simultaneously update the adversarial sample as well as network parameters, enabling the generation of strong adversaries during training, without additional computational overheads.

In order to mitigate gradient masking as seen in prior works that used single-step attacks for adversarial training, Vivek *et al.*[32] proposed the use of the R-MGM regularizer. The authors minimize the

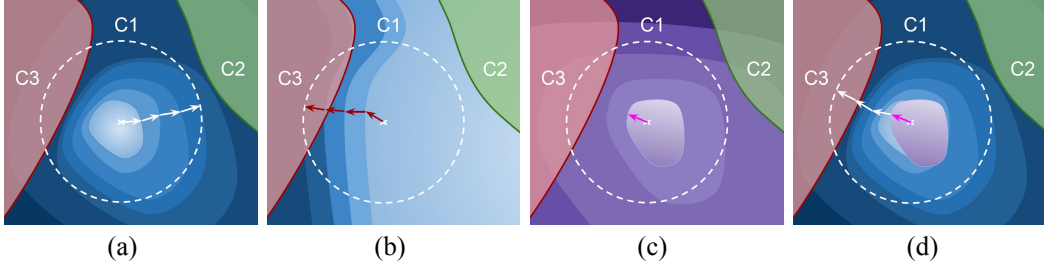

Figure 1: Schematic diagram of loss contours (a) Untargeted loss (b) Targeted loss w.r.t. class C3 (c) Guided loss for initial optimization (d) Path of adversary using GAMA

squared $\ell_2$ norm of the difference between logits corresponding to FGSM and R-FGSM adversaries to train adversarially robust models. In contrast to this, we introduce a regularizer to minimize the squared $\ell_2$ distance between the softmax outputs of clean and adversarial images, thereby improving the computational efficiency. Secondly, the adversary generation process uses the proposed Guided Adversarial Attack, thereby resulting in the use of a significantly stronger attack during training.

Contrary to prior wisdom, Wong *et al.*[34] (FBF), found the surprising result that R-FGSM training could indeed be successfully utilised to produce robust models. It was shown that R-FGSM adversarial training could be made effective with the use of small-step sizes for generation of adversaries, in combination with other techniques such as early-stopping and cyclic learning rates. With these techniques, they obtain better performance when compared to Adversarial Training for Free, with further reduction in computational requirements. While our proposed defense is also based on adversarial training with single-step adversaries, our choice of the loss function enables generation of stronger adversaries, thereby resulting in models that are significantly more robust. Further, we note that the acceleration techniques used in [34] can be utilized for our method as well.

## 4 Proposed Method

### 4.1 Impact of Initial Optimization Trajectory on Attack Efficacy

One of the most effective attacks known till date is the Projected Gradient Descent (PGD) attack [24], which starts with a random initialization and moves along the gradient direction to maximize cross entropy loss. Each iteration of PGD takes a step of a fixed size in the direction of sign of the gradient, after which the generated perturbation is projected back to the epsilon ball. Owing to the non-convex nature of the loss function, the initial gradient direction that maximizes cross-entropy loss may not lead to the optimal solution. This could lead to the given data sample being correctly classified, even if adversaries exist within an epsilon radius. This is shown in the schematic diagram of loss contours in Fig.1(a), where the adversary moves towards class C2 based on the initial gradient direction, and fails to find the adversary that belongs to class C3.

This is partly mitigated by the addition of initial random noise, which increases the chance of the adversary moving towards different directions. However, this gain can be seen only when the attack is run for multiple random restarts, thereby increasing the computational budget required for finding an adversarial perturbation. Another existing approach that gives a better initial direction to the adversaries is the replacement of the standard untargeted attack with a combination of multiple targeted attacks [15]. This diversifies the initial direction of adversaries over multiple random restarts, thereby resulting in a stronger attack. This can be seen in Fig.1(b), where the adversary is found by minimizing a targeted loss corresponding to the class C3, which has the closest decision boundary to the given sample. While this is a generic approach which can be used to strengthen any attack (including GAMA), it does not scale efficiently as the number of target classes increase.

In this paper, we propose to utilize supervision from the function mapping of clean samples in order to identify the initial direction that would lead to a stronger attack (Fig.1(c)). The proposed attack achieves an effect similar to the multi-targeted attack without having to explicitly minimize the loss corresponding to each class individually (Fig.1(d)). This leads to more reliable results in a single, or very few restarts of the attack, thereby improving the scalability of the attack to datasets with larger number of classes.

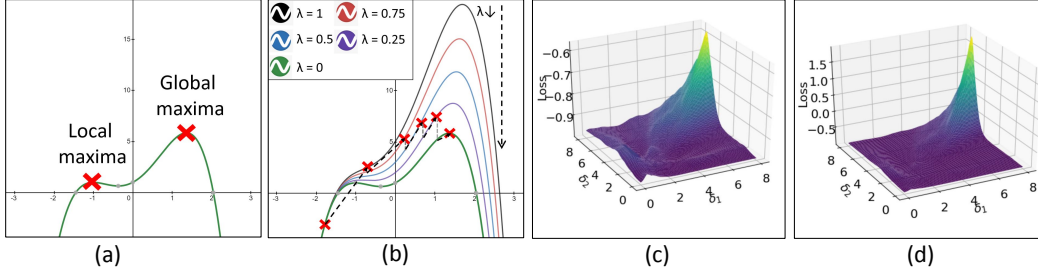

Figure 2: Addition of decaying $\ell_2$ relaxation term for function smoothing and improved optimization (a) 1-D example showing maximization of the non-concave function, $1 - x(x-2)(x+1)(x+1)$ (b) The smoothed function after addition of $\ell_2$ relaxation term is $1 - x(x-2)(x+1)(x+1) + \lambda(x+2)^2$. $\lambda$ is reduced from 1 to 0 over iterations. Optimization trajectory is shown using black dotted lines and red cross marks. (c, d) Plot of the loss surface of an FGSM trained model on perturbed images of the form $x^* = x + \delta_1 g + \delta_2 g^\perp$, obtained by varying $\delta_1$ and $\delta_2$. Here $g$ is the sign of the gradient direction of the loss with respect to the clean image $(x)$ and $g^\perp$ is a direction orthogonal to $g$. Loss functions used are: (c) Maximum-margin loss, and (d) GAMA loss as shown in Eq.1, with $\lambda$ set to 25. Addition of the relaxation term helps in smoothing the loss surface and suppressing gradient masking.

## 4.2 Guided Adversarial Margin Attack

Due to the inherent difficulty observed in the optimization of non-convex functions, several heuristic methods such as Graduated Optimization have been deployed to obtain solutions that sufficiently approximate global optima. To optimize a non-convex function, Graduated methods attempt to construct a family of smooth function approximations which are more amenable to standard optimization techniques. These function approximations are progressively refined in order to recover the original function toward the end of optimization. Hazan *et al.*[17] proposed to utilise projected gradient descent with a noisy gradient oracle, to optimize graduated function approximations obtained by local averaging over progressively shrinking norm balls. The authors characterise a family of functions for which their algorithm recovers approximate solutions of the global optima.

Along similar lines, we seek to introduce a relaxation term to obtain a series of smooth approximations of the primary objective function that is used to craft adversarial perturbations. We illustrate a simplified 1-dimensional example in Fig.2(a,b) to highlight the efficacy of graduated optimization through function smoothing. The loss function that is maximized for the generation of the proposed Guided Adversarial Margin Attack (GAMA) is as follows:

$$L = -f_\theta^y(\widetilde{x}) + \max_{j \neq y} f_\theta^j(\widetilde{x}) + \lambda \cdot ||\boldsymbol{f_\theta}(\widetilde{x}) - \boldsymbol{f_\theta}(x)||_2^2 \qquad (1)$$

The first two terms in the loss correspond to the maximum margin loss in probability space, which is the difference between the probability score of the true class $f_\theta^y(x)$, and the probability score of the second most confident class $j \neq y$. The standard formulation of PGD attack maximizes cross-entropy loss for the generation of attacks. We use maximum-margin loss here, as it is known to generate stronger attacks when compared to cross-entropy loss [5, 15]. In addition to this, we introduce a relaxation term corresponding to the squared $\ell_2$ distance between the probability vectors $f_\theta$ of the clean image $x$ and the perturbed image $\widetilde{x}$. This term is weighted by a factor $\lambda$ as shown in Eq.1. Similar to graduated optimization, this weighting factor is linearly decayed to 0 over iterations, so that this term only aids in the optimization process, and does not disturb the optimal solution of the true maximum-margin objective. As shown in Fig.2(c,d), the $\ell_2$ relaxation term indeed leads to a smoother loss surface in an FGSM trained model.

The gradients of this $\ell_2$ relaxation term are a weighted combination of the gradients of each of the class confidence scores of the perturbed image. Each term is weighted by the difference in corresponding class confidence scores of the perturbed image and clean image. Therefore, a direction corresponding to the gradient of a given class confidence score is given higher importance if it has already deviated by a large amount from the initial class confidence of the clean image. Thus, the weighting of the current gradient direction considers the cumulative effect of the previous steps, bringing about an advantageous effect similar to that of momentum. This helps direct the initial

---

**Algorithm 1** Guided Adversarial Margin Attack

---

1: **Input:** Network $f_\theta$ with parameters $\theta$, Input image $x$ with label $y$, Attack Size $\varepsilon$, Step-size $\eta$ or Convex parameter $\gamma$, Initial Weighting factor $\lambda_0$, Total Steps $T$, Relaxation Steps $\tau$, Learning Rate Schedule $S = \{t_1, \ldots, t_k\}$, Learning Rate Drop Factor $d$
2: **Output:** Adversarial Image $\widetilde{x}_T$
3: $\delta = Bern(-\varepsilon, \varepsilon)$        // Initialise Perturbation with Bernoulli Noise
4: $\widetilde{x}_0 = x_0 = x + \delta$ , $\lambda = \lambda_0$
5: **for** $t = 0$ **to** $T - 1$ **do**
6:     $L = \max_{j \neq y}\{f_\theta^j(\widetilde{x}_t)\} - f_\theta^y(\widetilde{x}_t) + \lambda \cdot ||f_\theta(\widetilde{x}_t) - f_\theta(x_0)||_2^2$
7:     $\lambda = \max(\lambda - \lambda_0/\tau, 0 )$
8:     **if** mode == PGD **then**
9:        $\delta = \delta + \eta \cdot sign(\nabla_\delta L)$
10:       $\delta = Clamp(\delta, -\varepsilon, \varepsilon)$       // Project Back To Constraint Set
11:    **else if** mode == FW **then**
12:       $\delta = (1 - \gamma) \cdot \delta + \gamma \cdot \varepsilon \cdot sign(\nabla_\delta L)$
13:    **end if**
14:     $\delta = Clamp(x + \delta, 0, 1) - x$
15:     $\widetilde{x}_{t+1} = x + \delta$
16:     **if** $t \in S$ **then**
17:       $\eta = \eta/d$ , $\gamma = \gamma/d$
18:    **end if**
19: **end for**

---

perturbation more strongly towards the class which maximizes the corresponding class confidence, while also making the optimization more robust to spurious random deviations due to local gradients.

The algorithm for the proposed attack is presented in Algorithm-1. The attack is initialized using random Bernoulli noise of magnitude $\varepsilon$. This provides a better initialization within the $\varepsilon$ bound when compared to Uniform or Gaussian noise, as the resultant image would be farther away from the clean image in this case, resulting in more reliable gradients initially. Secondly, the space of all sign gradient directions is represented completely by the vertices of the $\ell_\infty$ hypercube of a fixed radius around the clean image, which is uniformly explored using Bernoulli noise initialization. The attack is generated using an iterative process that runs over $T$ iterations, where the current step is denoted by $t$. At each step, the loss in Eq.1 is maximized to find the optimal $\widetilde{x}$ for the given iteration. The weighting factor $\lambda$ of the $\ell_2$ term in the loss function is linearly decayed to $0$ over $\tau$ steps.

We propose two variants of the Guided Adversarial Margin Attack, GAMA-PGD and GAMA-FW. GAMA-PGD uses Projected Gradient Descent for optimization, while GAMA-FW uses the Frank-Wolfe [13] algorithm, also known as Conditional Gradient Descent. In PGD, the constrained optimization problem is solved by first posing the same as an unconstrained optimization problem, and further projecting the solution onto the constraint set. Gradient ascent is performed by computing the sign of the gradient, and taking step of size $\eta$, after which the perturbation is clamped between $-\varepsilon$ and $\varepsilon$, to project to the $\ell_\infty$ ball. On the other hand, the Frank-Wolfe algorithm finds the optimal solution in the constraint set by iteratively updating the current solution as a convex combination of the present perturbation and the point within the constraint set that maximises the inner-product with the gradient. For the setting of $\ell_\infty$ constraints, this point which maximises the inner-product is simply given by epsilon times the sign of the current gradient. Since the constraint set is convex, this process ensures that the generated solution lies within the set, and hence does not require a re-projection to the same. This process results in a faster convergence, thereby resulting in stronger attacks when the budget for the number of iterations is small. This makes GAMA-FW particularly useful in the setting of adversarial training, where there is a fixed budget on the number of steps used for attack generation. Finally the image is clamped to be in the range $[0, 1]$. We use an initial step size of $\eta$ for GAMA-PGD and $\gamma$ for GAMA-FW, and decay this by a factor of $d$ at intermediate steps.

## 4.3 Guided Adversarial Training

In this section, we discuss details on the proposed defense GAT, which utilizes single-step adversaries generated using the proposed Guided Adversarial attack for training. As discussed in Section-4.2, the $\ell_2$ term between the probability vectors of clean and adversarial samples in Eq.1 provides reliable gradients for the optimization, thereby yielding stronger attacks in a single run. The effectiveness and

Table 1: **Attacks (CIFAR-10)**: Accuracy (%) of various defenses (rows) against adversaries generated using different 100-step attacks (columns) under the $\ell_\infty$ bound with $\varepsilon = 8/255$. Architecture of each defense is described in the column "Model". WideResNet is denoted by W (W-28-10 represents WideResNet-28-10), ResNet-18 is denoted by RN18, Pre-Act-ResNet-18 is denoted by PA-RN18. [‡]Additional data used for training, [†]$\varepsilon = 0.031$, [*]Defenses trained using single-step adversaries

| | Model | Single run of the attack | | | | | | 5 random restarts | | | | Top 5 targets | |
|---|---|---|---|---|---|---|---|---|---|---|---|---|---|
| | | PGD 100 | APGD CE | APGD DLR | FAB | GAMA PGD | GAMA FW | APGD DLR | FAB | GAMA PGD | GAMA FW | MT | GAMA PGD-MT |
| Carmon *et al.*[7][‡] | W-28-10 | 61.86 | 61.81 | 60.85 | 60.88 | **59.81** | 59.83 | 60.64 | 60.62 | **59.65** | 59.71 | 59.86 | **59.56** |
| Sehwag *et al.*[27][‡] | W-28-10 | 59.93 | 59.61 | 58.39 | 58.29 | 57.51 | **57.50** | 58.26 | 58.06 | **57.37** | 57.38 | 57.48 | **57.20** |
| Wang *et al.*[33][‡] | RN18 | 52.87 | 52.38 | 49.70 | 48.50 | **48.12** | 48.17 | 49.37 | 48.33 | **47.92** | 47.97 | 47.76 | **47.58** |
| Wang *et al.*[33][‡] | W-28-10 | 62.63 | 61.76 | 58.98 | 57.53 | 57.19 | **57.14** | 58.56 | 57.29 | **56.84** | 56.92 | 56.80 | **56.54** |
| Hendrycks *et al.*[19][‡] | W-28-10 | 57.58 | 57.20 | 57.25 | 55.55 | 55.24 | **55.19** | 56.96 | 55.40 | 55.11 | **55.08** | 55.06 | **54.92** |
| Rice *et al.*[26] | W-34-20 | 57.25 | 56.93 | 55.99 | 54.34 | **53.77** | 53.88 | 55.70 | 54.19 | **53.64** | 53.68 | 53.59 | **53.45** |
| Zhang *et al.*[37][†] | W-34-10 | 55.60 | 55.30 | 54.18 | 53.92 | **53.29** | 53.38 | 54.04 | 53.82 | **53.17** | 53.22 | 53.32 | **53.09** |
| Madry *et al.*[24] [12] | RN-50 | 53.49 | 51.78 | 53.03 | 50.67 | **50.04** | 50.08 | 52.64 | 50.37 | **49.81** | 49.92 | 49.76 | **49.41** |
| Wong *et al.*[34][*] | PA-RN18 | 46.42 | 45.96 | 46.95 | 44.51 | **43.85** | 43.90 | 46.64 | 44.03 | **43.65** | 43.69 | 43.65 | **43.33** |
| GAT (Ours)[*] | W-34-10 | 55.10 | 54.73 | 53.08 | 51.28 | **50.76** | 50.79 | 52.75 | 51.07 | **50.43** | 50.48 | 50.45 | **50.18** |

efficiency of the proposed attack make it suitable for use in adversarial training, to generate more robust defenses. This attack is notably more useful for training single-step defenses, where reliance on the initial direction is significantly higher when compared to multi-step attacks.

Initially, Bernoulli noise of magnitude $\alpha$ is added to the input image in order to overcome any possible gradient masking effect in the vicinity of the data sample. Next, an attack is generated by maximizing loss using single step optimization. We use the minimax formulation proposed by Madry *et al.*[24] for adversarial training, where the maximization of a given loss is used for the generation of attacks, and minimization of the same loss on the generated adversaries leads to improved robustness. In order to use the same loss for both attack generation and training, we use cross-entropy loss instead of the maximum-margin loss in Eq.1. This improves the training process, as cross-entropy loss is known to be a better objective for training when compared to maximum-margin loss. The generated perturbation is then projected onto the $\varepsilon$-ball. We introduce diversity in the generated adversaries by setting $\lambda$ to 0 in alternate iterations, only for the attack generation. These adversarial samples ($\widetilde{x_i}$) along with the clean samples ($x_i$) are used for adversarial training. The algorithm of the proposed single-step defense GAT is presented in detail in Algorithm-1 of the Supplementary section.

Single-step adversarial training methods commonly suffer from gradient masking, which prevents the generation of strong adversaries, thereby leading to weaker defenses. The proposed training regime caters to the dual objective of minimizing loss on adversarial samples, while also explicitly enforcing function smoothing in the vicinity of each data sample (Details in Section-1 of the Supplementary section). The latter outcome strengthens the credibility of the linearity assumption used during generation of single-step adversaries, thereby improving the efficacy of the same. This coupled with the use of stronger adversaries generated using GAMA enables GAT to achieve state-of-the-art robustness among the single-step training methods.

## 5 Experiments and Analysis

In this section, we present details related to the experiments conducted to validate our proposed approach. We first present the experimental results of the proposed attack GAMA, followed by details on evaluation of the proposed defense GAT. The primary dataset used for all our evaluations is CIFAR-10 [20]. We also show results on MNIST [23] and ImageNet [11] for the proposed attack GAMA in the main paper and for the proposed defense GAT in Section-6 of the Supplementary. We use the constraint set given by the $\ell_\infty$ ball of radius $8/255$, $8/255$ and 0.3 for the CIFAR-10, ImageNet and MNIST datasets respectively. The implementation details of the proposed attack and defense are presented in Sections-3 and 4 of the Supplementary.

### 5.1 Evaluation of the proposed attack (GAMA)

The performance of various defenses against different attack methods on CIFAR-10 dataset is shown in Table-1. We present results for both a single run of the attack (with a budget of 100 iterations), as well as the worst-case accuracy across 5 random restarts (with an effective budget of $5 \times 100$

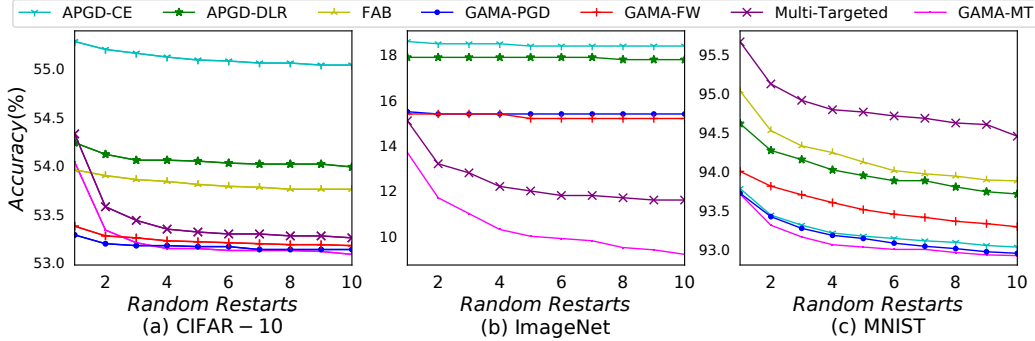

Figure 3: Accuracy (%) of different attacks against multiple random restarts. Evaluations are performed on TRADES WideResNet-34 model [37] for CIFAR-10, Madry *et al.*[12] ResNet-50 model for ImageNet (first 1000 samples), and TRADES SmallCNN [37] model for MNIST.

iterations). Notably, GAMA-PGD and GAMA-FW consistently outperform all other untargeted attacks across all defenses. Further, we remark that while the FAB attack stands as the runner-up method, it requires significantly more computation time, approximately 6 times that of GAMA-PGD and GAMA-FW.

The Multi-Targeted attack (MT) is performed by targeting the top 5 classes excluding the correct class. We present GAMA-MT, a multi-targeted version of the GAMA-PGD attack, where the maximum-margin loss is replaced by the margin loss targeted towards the top 5 classes excluding the true class. We note that the GAMA-MT attack is consistently the most effective attack across all defenses.

We further present evaluations on the TRADES WideResNet-34 model [37], PGD adversarially trained ResNet-50 model [24] and TRADES SmallCNN model [37] on the CIFAR-10, ImageNet (first 1000 samples) and MNIST datasets respectively against different attack methods in Fig.3. We find that while GAMA-PGD and GAMA-FW continue to consistently achieve the strongest attacks, they are also less sensitive to the random initialization, when compared to other attack methods for varying number of random restarts. Thus the proposed attacks offer a more reliable bound on the robustness of models, within a single restart or very few restarts. The proposed multi-targeted attack GAMA-MT outperforms all other attacks significantly on ImageNet, and is marginally better than GAMA-PGD for CIFAR-10 and MNIST.

We evaluate the proposed attack on the TRADES leaderboard models [37]. A multi-targeted version of our attack GAMA on the WideResNet-34 CIFAR-10 model achieved the top position in the leaderboard, with 53.01% for a 100-step attack with 20 random restarts. On the SmallCNN MNIST model, we achieve an accuracy of 92.57% for a 100-step attack with 1000 random restarts.

**Ablation Experiments:** We present evaluations on the TRADES WideResNet-34 model on the CIFAR-10 test set with several ablations of the proposed attack in Table-2 of the Supplementary section. We first observe that the maximum-margin loss is more effective when compared to the cross-entropy loss, for both 10 and 100 step attacks. Further, we observe that we obtain stronger adversaries while optimizing the margin loss between predicted probability scores, as compared to the corresponding logits. The weighting factor for the squared $\ell_2$ relaxation term is linearly decreased to 0 for the 100-step attack, while it is kept constant for the 10-step attack. From the 100-step evaluations, we observe that the graduated optimization indeed aids in finding stronger adversaries. Further, the addition of initial Bernoulli random noise aids in improving 100-step adversaries. We also note that GAMA-FW achieves the strongest attack when the available budget on the number of steps for attack is relatively small.

## 5.2 Evaluation of the proposed defense (GAT)

The white-box accuracy of the proposed defense GAT is compared with existing defenses in Table-2. In addition to evaluation against standard attacks, we also report accuracy on the recently proposed ensemble of attacks called AutoAttack [10], which has been successful in bringing down the accuracy of many existing defenses by large margins. The existing single-step defenses are presented in the first partition of the table and the multi-step defenses are presented in the second. The proposed

Table 2: **Defenses (CIFAR-10)**: Accuracy (%) of different models (rows) against various $\ell_\infty$ norm bound ($\varepsilon = 8/255$, $^\dagger \varepsilon = 0.031$) white-box attacks (columns). The first partition corresponds to single-step defenses, and the second has multi-step defenses. For the C&W attack, the mean $\ell_2$ norm required to achieve high Fooling Rate (FR) is reported. Higher the $\ell_2$ norm, better is the robustness.

| Method | Model | Clean Acc (%) | FGSM | IFGSM 7-step | PGD (n-steps) 7 | PGD (n-steps) 20 | PGD (n-steps) 500 | GAMA PGD-100 | AA | C & W Mean $\ell_2$ |
|---|---|---|---|---|---|---|---|---|---|---|
| Normal | RN18 | 92.30 | 15.98 | 0.00 | 0.00 | 0.00 | 0.00 | 0.00 | 0.00 | 0.108 |
| FGSM-AT [14] | RN18 | 92.89 | **96.94** | 0.82 | 0.38 | 0.00 | 0.00 | 0.00 | 0.00 | 0.078 |
| RFGSM-AT [31] | RN18 | 89.24 | 49.94 | 42.52 | 41.02 | 35.02 | 34.17 | 33.87 | 33.16 | 0.634 |
| ATF [28] | RN18 | 71.77 | 46.67 | 45.06 | 44.96 | 43.53 | 43.52 | 40.34 | 40.22 | 0.669 |
| FBF [34] | RN18 | 82.83 | 54.09 | 50.28 | 49.66 | 46.41 | 46.03 | 43.85 | 43.12 | 0.685 |
| R-MGM [32] | RN18 | 82.29 | 55.04 | 50.87 | 50.03 | 46.23 | 45.79 | 44.06 | 43.72 | 0.745 |
| GAT (**Ours**) | RN18 | 80.49 | 57.37 | 55.32 | 54.99 | 53.13 | 53.08 | 47.76 | 47.30 | **0.762** |
| FBF [34] | WRN34 | 82.05 | 53.79 | 49.20 | 49.51 | 46.35 | 45.94 | 43.13 | 43.14 | 0.628 |
| GAT (**Ours**) | WRN34 | 85.17 | 61.93 | **58.68** | **57.25** | **55.34** | **55.10** | 50.76 | **50.27** | 0.724 |
| PGD-AT [24] | RN18 | 82.67 | 54.60 | 51.15 | 50.38 | 47.35 | 46.96 | 44.94 | 44.57 | 0.697 |
| TRADES [37] | RN18 | 81.73 | 57.39 | 54.80 | 54.43 | 52.39 | 52.16 | 48.95 | 48.75 | 0.743 |
| TR-GAT (**Ours**) | RN18 | 81.32 | 57.61 | 55.34 | 55.13 | 53.37 | 53.22 | 49.77 | 49.62 | **0.744** |
| TRADES [37] $^\dagger$ | WRN34 | 84.92 | 61.06 | 58.47 | 58.09 | 55.79 | 55.56 | 53.29 | 53.18 | 0.705 |
| TR-GAT (**Ours**) | WRN34 | 83.58 | **61.22** | **58.69** | **58.98** | **57.07** | **56.89** | **53.43** | **53.32** | 0.719 |

single-step defense GAT outperforms the current state-of-the-art single-step defense, FBF [34] on both ResNet-18 [18] and WideResNet-34-10 [36] models by a significant margin. In fact, we find that increasing model capacity does not result in an increase in robustness for FBF due to catastrophic overfitting. However, with the proposed GAT defense, we obtain a $2.97\%$ increase in worst-case robust accuracy by using a larger capacity model, alongside a significant boost of $4.68\%$ in clean accuracy. In addition to these results, the GAT WideResNet-34-10 model is also evaluated against other state-of-the-art attacks, including our proposed attack GAMA in Table-1. Here, GAMA also serves as an adaptive attack to our defense, as the same loss formulation is used for both. We present evaluations on black-box attacks, gradient-free attacks, targeted attacks, untargeted attacks with random restarts and more adaptive attacks Section-6 of the Supplementary. We also present all the necessary evaluations to ensure the absence of gradient masking [3] in the Supplementary material.

We further analyse the impact of using the proposed Guided Adversarial attack for adversary generation in the TRADES training algorithm. We utilize adversaries generated using GAMA-FW for this, as this algorithm generates stronger 10-step attacks when compared to others. Using this approach, we observe marginal improvement over TRADES accuracy. This improves further by replacing the standard adversaries used for TRADES training with GAMA-FW samples only in alternate iterations. We present results on the proposed 10-step defense TR-GAT using this combined approach in Table-2.

The improvement in robustness with the use of Guided Adversarial attack based adversaries during training is significantly larger in single-step adversarial training when compared to multi-step adversarial training. This is primarily because single-step adversarial training is limited by the strength of the adversaries used during training, while the current bottleneck in multi-step adversarial training methods is the amount of data available for training [7].

# 6    Conclusions

We propose Guided Adversarial Margin Attack (GAMA), which utilizes the function mapping of clean samples to guide the generation of adversaries, resulting in a stronger attack. We introduce an $\ell_2$ relaxation term for smoothing the loss surface initially, and further reduce the weight of this term gradually over iterations for better optimization. We demonstrate that our attack is consistently stronger than existing attacks across multiple defenses. We further propose to use Frank-Wolfe optimization to achieve faster convergence in attack generation, which results in significantly stronger 10-step attacks. We utilize the adversaries thus generated to achieve an improvement over the current state-of-the-art adversarial training method TRADES. The proposed Guided Adversarial attack aids the initial steps of optimization significantly, thereby making it suitable for single-step adversarial training. We propose a single-step defense, Guided Adversarial Training (GAT) which uses the proposed $\ell_2$ relaxation term for both attack generation and adversarial training, thereby achieving a significant improvement in robustness over existing single-step adversarial training methods.

# 7 Broader Impact

As Deep Networks see increasing utility in everyday life, it is essential to be cognizant of their worst-case performance and failure modes. Adversarial attacks in particular could have disastrous consequences for safety critical applications such as autonomous navigation, surveillance systems and medical diagnosis. In this paper, we propose a novel adversarial attack method, GAMA, that reliably bounds the worst-case performance of Deep Networks for a relatively small computational budget. We also introduce a complementary adversarial training mechanism, GAT, that produces adversarially robust models while utilising only single-step adversaries that are relatively cheap to generate. Thus, our work has immense potential to have a positive impact on society, by enabling the deployment of adversarially robust Deep Networks that can be trained with minimal computational overhead. During the development phase of systems that use Deep Networks, the GAMA attack can be used to provide reliable worst-case evaluations, helping ensure that systems behave as expected when deployed in real-world settings. On the negative side, a bad-actor could potentially use the proposed attack to compromise Deep Learning systems. However, since the proposed method is a white-box attack, it is applicable only when the entire network architecture and parameters are known to the adversary, which is a relatively rare scenario as model weights are often kept highly confidential in practice.

# 8 Acknowledgments and Disclosure of Funding

This work was supported by Uchhatar Avishkar Yojana (UAY) project (IISC_10), MHRD, Govt. of India. We would like to extend our gratitude to all the reviewers for their valuable suggestions.

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
