[Supplementary Material]

# Supplementary material
# Guided Adversarial Attack for Evaluating and Enhancing Adversarial Defenses

**Gaurang Sriramanan**\*, **Sravanti Addepalli**\*, **Arya Baburaj**, **R.Venkatesh Babu**
Video Analytics Lab, Department of Computational and Data Sciences
Indian Institute of Science, Bangalore, India

## 1 Improved local properties induced by Guided Adversarial Training (GAT)

In this section, we present details on the improved local properties achieved using the proposed single-step defense, GAT (Guided Adversarial Training).

We examine the local properties of networks trained using the proposed methodology here. Formally, a function $f$ is locally Lipschitz on a metric space $\mathcal{X}$, if for every $x \in \mathcal{X}$, there exists a neighborhood $U(x)$ such that $f$ restricted to $U(x)$ is Lipschitz continuous, that is,

$$\|f(x) - f(y)\| \leq \mathcal{L} \cdot \|x - y\| \qquad \forall y \in U(x) \tag{1}$$

In our framework, we consider $\mathcal{X}$ to be the data-manifold, and $f_\theta$ as the softmax output of the neural network, where $\theta$ represents the parameters of the network. We first study the impact of the proposed squared $\ell_2$ distance term in the loss function. Minimisation of this regularizer term leads to the following solution for $\theta^*$:

$$\theta^* = \underset{\theta}{\operatorname{argmin}}\|f_\theta(x) - f_\theta(\widetilde{x})\|_2^2 \tag{2}$$

where $\widetilde{x}$ is an adversary corresponding to clean image $x$. We note that the gradient of the loss on $f_\theta(x)$ represents the direction of steepest increase of the loss function. Thus, given that we want to obtain the strongest adversary achievable within a single backward-pass of the loss, we find $\widetilde{x}$ as given in Alg.1, L6 to L9.

Since we want the network to be robust to adversaries lying within the $\ell_\infty$-ball of radius $\varepsilon$ centered at $x$, we ideally want a function $f_{\theta^*}$ that is locally Lipschitz within $U_\varepsilon(x)$:

$$U_\varepsilon(x) = \{x' : \|x - x'\|_\infty \leq \varepsilon\} \tag{3}$$

Given that $\|x - \widetilde{x}\|_\infty \leq \varepsilon$, we have,

$$\|x - \widetilde{x}\|_2 = \sqrt{\sum_{i=1}^{d}(x_i - \widetilde{x}_i)^2} \leq \sqrt{\sum_{i=1}^{d}\varepsilon^2} \leq \sqrt{d} \cdot \varepsilon \tag{4}$$

where $d$ is the dimension of the input space. For constrained adversaries, we now have,

$$\|f_{\theta^*}(x) - f_{\theta^*}(\widetilde{x})\|_2 \leq \sqrt{d} \cdot \varepsilon \cdot \mathcal{L} \tag{5}$$

Thus, the prediction of $f_{\theta^*}$ is constant on $U_\varepsilon(x)$, if the Lipschitz constant $\mathcal{L}$ is sufficiently small. Note that under this (strong) assumption, $f_{\theta^*}$ is *guaranteed* to be adversarially robust, as it predicts the same class for all images in the $\varepsilon$-constraint.

---

---

**Algorithm 1** Guided Adversarial Training

---

1: **Input:** Network $f_\theta$ with parameters $\theta$, Training Data $\mathcal{D} = \{(x_i, y_i)\}$, Minibatch Size M, Attack Size $\varepsilon$, Initial Noise Magnitude $\alpha$, Epochs $E$, Learning Rate $\eta$
2: **for** $epoch = 1$ **to** $E$ **do**
3:   **for** minibatch $B_j \subset \mathcal{D}$ **do**
4:     Set $L = 0$
5:     **for** $i = 1$ **to** $M$ **do**
6:       $\delta = Bern(-\alpha, \alpha)$
7:       $\delta = \delta + \varepsilon \cdot \text{sign}\big(\nabla_\delta \big(\ell_{CE}(f_\theta(x_i + \delta), y_i) + \lambda \cdot ||f_\theta(x_i + \delta) - f_\theta(x_i)||_2^2\big)\big)$
8:       $\delta = Clamp\,(\delta, -\varepsilon, \varepsilon)$
9:       $\widetilde{x}_i = Clamp\,(x_i + \delta, 0, 1)$
10:      $L = L + \ell_{CE}(f_\theta(x_i), y_i) + \lambda \cdot ||f_\theta(\widetilde{x}_i) - f_\theta(x_i)||_2^2$
11:     **end for**
12:     $\theta = \theta - \frac{1}{M} \cdot \eta \cdot \nabla_\theta L$
13:   **end for**
14: **end for**

---

We note that the set $\mathcal{X}$ is compact, since it is a closed and bounded subset of $[0, 1]^d$. Since the function $f_\theta$ is differentiable over $\mathcal{X}$, it is uniformly continuous over $\mathcal{X}$ as well. Thus, there exists $\varepsilon' > 0$, such that prediction of $f_\theta$ is constant on $U_{\varepsilon'}(x)$ for all $x \in \mathcal{X}$. Thus, since adversarial perturbations cannot lie within an $\varepsilon'$-ball of any sample $x$, we are primarily interested in adversaries $\widetilde{x}$ such that:

$$\widetilde{x} \in \mathcal{A}(x) = \{x' : \varepsilon' < \|x - x'\|_\infty \leq \varepsilon, f_\theta(x) \neq f_\theta(x')\} \tag{6}$$

Thus, the local Lipschitz constant $\mathcal{L}$ of interest is given by:

$$\mathcal{L} = \sup_{\widetilde{x} \in \mathcal{A}(x)} \frac{\|f_\theta(x) - f_\theta(\widetilde{x})\|_2}{\|x - \widetilde{x}\|_2} < \sup_{\widetilde{x} \in \mathcal{A}(x)} \frac{1}{\sqrt{d} \cdot \varepsilon'} \|f_\theta(x) - f_\theta(\widetilde{x})\|_2 \tag{7}$$

The square of the expression on the RHS is precisely the regularisation term used in the proposed loss function for training. Hence, imposing the proposed regularizer encourages the optimization procedure to produce a network that is locally Lipschitz continuous, with a smaller local Lipschitz constant. The actual optimisation procedure minimises the combined loss, with the first term given by the cross-entropy term, and the squared $\ell_2$ loss term weighted by a factor $\lambda$. Note that without the inclusion of the first term, several degenerate solutions are possible, for example, a network that is constant for all images $x$. The value of $\lambda$ determines the *effective* learning rate for the squared $\ell_2$ loss term, and thus enforces the extent of function smoothness (refer Section-3.2). The value of $\lambda$ can be chosen so as to achieve the desired trade-off between clean accuracy and robustness [16]. For a fixed $\lambda$, we obtain a family of functions $\mathcal{F}_\lambda$ that achieve the same cross-entropy loss. We can then extend the same analysis to $f_{\theta*}$ which is the minimiser of the squared $\ell_2$ loss term, and thus the combined total loss, amongst all functions $f_\theta \in \mathcal{F}_\lambda$.

## 2 Details on the datasets used

We run extensive evaluations on MNIST [10], CIFAR-10 [9] and ImageNet [5] datasets to validate our claims on the proposed attack and defense.

MNIST [10] is a handwritten digit recognition dataset consisting of 60,000 training images and 10,000 test images. The images are grayscale, and of dimension 28×28. We split the training set into a random subset of 50,000 training images and 10,000 validation images.

CIFAR-10 [9] is a popular dataset in computer vision research, consisting of the following ten classes: Airplane, Automobile, Bird, Cat, Deer, Dog, Frog, Horse, Ship and Truck. The similarity of classes such as Cat and Dog make this a challenging dataset for the domain of adversarial robustness. The dimension of each image in this dataset is $32 \times 32 \times 3$. The original training set comprises of 50,000 images which we split into 49,000 training images and 1,000 validation images (equally balanced across all ten classes), while the test set has 10,000 images.

ImageNet [5] is a 1000-class dataset consisting of approximately 1.2 million training images and 50,000 images in the validation set. This dataset has a private test set which is not available for access

Table 1: Architecture used for MNIST dataset. Modified LeNet architecture is used for training robust models. We use the network, BB-MNIST as the source model for Black-Box attacks on the MNIST dataset

| Modified LeNet (M-LeNet) | BB-MNIST |
|---|---|
| {conv(32,5,5) + Relu}×2 | Conv(64,5,5) + Relu |
| MaxPool(2,2) | Conv(64,5,5) + Relu |
| {conv(64,5,5) + Relu}×2 | Dropout(0.25) |
| MaxPool(2,2) | FC(128) + Relu |
| FC(512) + Relu | Dropout(0.5) |
| FC + Softmax | FC + Softmax |

to the public. Therefore, we use the designated validation set as the test set for our experiments. Furthermore, we split the designated training set into an 80-20 train-validation split. For training and evaluation of our proposed defense, we consider a random 100-class subset of this dataset, in order to ease the computation time and resource requirements. Even this subset is challenging due to the large dimensionality of the input space ($224 \times 224 \times 3$) and the high level of similarity between different classes. The set of 100 classes used for our experiments is shared along with our codes.

We use NVIDIA DGX workstation with V100 GPUs for our training and evaluations. The proposed single-step defense takes approximately 2 hours for training on ResNet-18 architecture for CIFAR-10 dataset. The proposed 100-step GAMA-PGD attack takes approximately 5 minutes for a single run to evaluate a ResNet-18 model on the CIFAR-10 test set.

# 3 Details on Guided Adversarial Training

In this section, we present implementation details of the proposed defense.

## 3.1 Architecture details

For evaluation of the proposed defense, we select a fixed architecture for each dataset, and use the same architecture to report results across all existing defense methods as well. We use a modified LeNet architecture with 4 convolutional layers as shown in Table-1 for MNIST, and the ResNet-18 [7] architecture for our experiments on CIFAR-10 and ImageNet-100 datasets. We also report results on WideResNet-34-10 [20] architecture for the CIFAR-10 dataset with the proposed defense.

## 3.2 Details on the training algorithm

The training algorithm for the proposed single-step defense GAT is presented in Algorithm-1. This is explained in Section-4.3 of the main paper. In the proposed defense GAT, we first add Bernoulli noise of magnitude $\alpha$ to the clean image. We set the value of $\alpha$ to be either $\varepsilon$ or $\varepsilon/2$ [15]. In the next step, we generate a GA-CE (Guided Adversarial Cross-Entropy) attack on the noise-added image, and finally project the generated perturbation onto the $\varepsilon$-ball of the clean image. In standard single-step adversarial training [15], the FGSM attack is of magnitude ($\varepsilon - \alpha$). So, the perturbation is always within the $\ell_\infty$ $\varepsilon$-ball of the clean image and thus there is no need to project it back to the $\varepsilon$-ball. The proposed formulation however, can move the adversary outside the $\varepsilon$-ball thereby increasing the likelihood of the adversary lying on the boundary of the $\varepsilon$-ball after projection. In principle, this broader class of adversaries should lead to a stronger attack, since most of the adversaries are farther away from the original image when compared to a standard R-FGSM attack.

The two losses in the proposed framework (Alg.1, L7,10) are combined by weighting the squared $\ell_2$ loss term by a factor $\lambda$. This weighting term determines the trade-off between accuracy and robustness [16] as shown in Fig.1(b). We observe that during the initial stages of training, the loss surface is relatively smooth, thereby requiring a low value for $\lambda$. This factor is stepped up towards the end of training. The learning-rate is decayed when the loss begins to plateau. We generally observe that for the first learning-rate update, both clean and adversarial accuracy improve in tandem if $\lambda$ is kept fixed. As training progresses, the loss surface becomes increasingly convoluted; we thus step-up

Figure 1: (a) Accuracy (%) of the TRADES-WRN34 model [21] trained on CIFAR-10 dataset, against the proposed GAMA-PGD and GAMA-FW attacks, across various settings of $\lambda$ in Eq.1 of the main paper. This is the weight of $\ell_2$ relaxation term in the loss. (b, c) Accuracy (%) of the proposed GAT defense on ResNet-18 models trained on CIFAR-10 dataset, across variation in hyperparameters used for the defense. $\lambda$ (Alg.-1, L7, 10) is varied in (b) and $\lambda$ step up factor is varied in (c). Accuracy (%) on clean samples and 100-step GAMA-PGD adversaries is shown. GAMA-PGD attack settings are fixed to optimal values.

Figure 2: Plot of Cross-Entropy loss on CIFAR-10 test samples across epochs for different training methods: (a) R-FGSM-AT (b) PGD-AT (c) TRADES (d) GAT (**Ours**)

$\lambda$ along with the subsequent decay in learning rate, so as to strike a balance between clean accuracy and adversarial robustness.

The $\lambda$ step-up factor can be viewed as an effective learning rate increase for the squared $\ell_2$ loss term. During adversarial training involving step learning rate decay, accuracy is boosted significantly for the initial few step decays. This results in a change in loss landscape, leading to an increase in loss on adversarial samples, as can be seen in case of R-FGSM training and PGD training in Fig. 2 (a and b). The training on adversarial samples is however unable to compensate for the increase in loss as the learning rate is too low. Thus, inclusion of the step-up factor ensures that the adversarial loss does not increase rapidly over epochs. It can be seen in Fig.2 (d) that a combination of the proposed learning rate schedule and step-up factor is able to prevent an increase in loss. Similar to TRADES, the loss on clean and adversarial samples consistently reduces over epochs in the proposed method. It is also worth noting that the loss on FGSM samples is very close to the loss on PGD samples, thereby proving the effectiveness of training with single-step adversaries generated using the Guided Adversarial attack.

### 3.3 Implementation details

**GAT (Single-step defense):** Before generating the attack for adversarial training in GAT defense, we add initial noise of magnitude $\varepsilon$ for MNIST, and $\varepsilon/2$ for the other datasets. For the CIFAR-10 dataset, we use an initial $\lambda$ and step-up factor of 10 and 4 respectively for ResNet-18 training, and 3 and 20 respectively for WideResNet-34-10 training. The same values of $\lambda$ and step-up factor are used for both generation of attack (Alg.1, L7) and training (Alg.1, L10). We use the SGD optimizer with momentum of 0.9 and weight decay of 5e-4 for all our experiments. The learning rate is set to 0.1 and decayed by a factor of 10, at epochs 70 and 85 for ResNet-18 and at epochs 55,70 and 75 for WideResNet-34-10. As discussed in Section-3.2, the second learning rate update is accompanied by a $\lambda$ step-up. The impact of variation in $\lambda$ and $\lambda$ step-up factor is shown in Fig.1(b) and Fig.1(c) respectively. It can be observed that the clean accuracy and accuracy on GAMA-PGD adversarial samples is stable across variations in the hyperparameters. As $\lambda$ increases, there is a reduction in

clean accuracy and an increase in robustness. This trend continues till a value of 10, after which the adversarial accuracy remains constant or starts reducing. We therefore select 10 as the optimum value. A similar trend is observed for variation in $\lambda$ step-up factor as well. The slight reduction in robustness at high values of $\lambda$ or $\lambda$ step-up factor is due to the reduction in clean accuracy. Since accuracy on clean samples is an upper bound on adversarial robustness, over-regularization causes both to reduce.

For ImageNet-100, we use an initial $\lambda$ of 20 and step it up by a factor of 7 at epoch 90. The learning rate is 0.1 initially and decayed by a factor of 10 at epochs 60 and 90.

For MNIST, the initial learning rate is set to 0.01, and further decayed by a factor of 5 three times at regular intervals. In MNIST dataset, the clean accuracy shoots up to above 90% within the first epoch, and reaches a very high value in a few epochs. Hence, we do not need a simple drop in learning rate without $\lambda$ step-up, for a further increase in robust accuracy. We therefore include the $\lambda$ step-up factor at all three times of learning rate update. We set the value of $\lambda$ to 15 and the $\lambda$ step-up factor to 3.

We train our MNIST model for 50 epochs, CIFAR-10 model for 100 epochs and ImageNet-100 model for 120 epochs. The other methods are trained until convergence.

**TR-GAT (Multi-step defense):** We next present details on the proposed multi-step defense TR-GAT. To incorporate Guided Adversarial Attacks for TRADES training as shown in Table-2 in the main paper, we alternate between a 10-step PGD attack that maximises KL-Divergence and a 10-step GAMA-FW attack with a constant $\lambda$, set to the same value as the weighting used for the KL-Divergence term in training. We set this weighting term (called $\beta$ in the TRADES paper [21]) to be 5 and 6 respectively for the ResNet-18 and WideResNet-34 models on CIFAR-10 dataset. For training the TR-GAT model, we use the same learning rate schedule and total epochs as used for the corresponding TRADES model.

## 4   Implementation details of the Guided Adversarial Margin Attack

The loss function that is maximized for generation of our proposed attack GAMA is shown in Eq.1 of the main paper. This consists of two terms, maximum margin loss and squared $\ell_2$ relaxation term between the softmax vectors of clean and perturbed images. The squared $\ell_2$ term is weighted by a factor $\lambda$, which is decayed to 0 over a fixed number of iterations. We set $\lambda$ to 50 and decay this to 0 over 25 iterations. This aids the optimization process by providing a better initial direction. As shown in Fig.1(a), the attack strength is stable over a wide range of $\lambda$ values.

Prior to the attack generation, we add Bernoulli noise of magnitude $\varepsilon$ to the image. Analogous to the training of Deep Neural Networks, generation of standard attacks are also known to benefit with a step learning rate schedule over the optimization process [6]. For the GAMA-PGD attack, we use an initial step size of $2 \cdot \varepsilon$ and decay it by a factor of 10 at iterations 60 and 85 for a 100-step schedule. Similarly, for the GAMA-FW attack, we use an initial $\gamma$ of 0.5 and decay it by a factor of 5 at the same iterations. For evaluating the TRADES leaderboard [21] WideResNet-34 CIFAR-10 model, we use a multi-targeted version of our attack; we run the attack for 100 steps, and 20 random restarts, wherein we alternate between the proposed GAMA loss and the margin loss corresponding to different classes over multiple restarts.

As noted by Gowal et al. [6], the loss surface of models adversarially trained on the MNIST datasets is complex. This necessitates different attack settings for this dataset. The threat model considered typically for MNIST is $\varepsilon = 0.3$. We set the initial $\lambda$ to 5 and decay it to 0 in 50 iterations. For GAMA-PGD, we use an initial step size of $\varepsilon$ and decay it by 10 at iterations 50 and 75. For GAMA-FW, $\gamma$ of 0.5 is used initially and decayed by a factor of 5 at the same iterations.

## 5   Details on Evaluation of the proposed attack

For evaluation of the proposed attacks (Table-1 and Fig.3 of the main paper, Fig.1(a) of the supplementary), we use pre-trained models shared by the respective authors of various defenses. Therefore, the architecture of different models would be as chosen by the respective authors, and is thus not consistent across all defenses presented in Table-1 of the main paper.

Table 2: **GAMA (Attack) Ablations (CIFAR-10)**: Accuracy (%) of various attacks (rows) on TRADES WRN-34-10 [21] model under the $\ell_\infty$ bound with $\varepsilon = 0.031$, across different settings of number of steps and restarts.

| Attacks | 100 - step attacks | | 10 - step attacks | |
|---|---|---|---|---|
| | Single run | 5 restarts | Single run | 5 restarts |
| PGD (Cross-entropy loss) | 55.61 | 55.27 | 56.7 | 56.31 |
| PGD (Margin loss in logits space) | 54.19 | 54.07 | 55.04 | 54.75 |
| PGD (Margin loss in prob. space) | 53.94 | 53.8 | 54.87 | 54.68 |
| PGD (Margin loss and $\ell_2$ loss in prob. space) | 53.73 | 53.54 | 54.96 | 54.67 |
| GAMA - PGD | **53.29** | **53.17** | 54.95 | 54.66 |
| GAMA - FW | 53.38 | 53.22 | **54.27** | **54.00** |

## 5.1 Ablation Experiments

We present evaluations on the TRADES WideResNet-34 model on the CIFAR-10 test set with several ablations of the proposed attack in Table-2. We first observe that the maximum-margin loss, which is similar to the C&W $\ell_\infty$ based attack [2], is more effective when compared to the cross-entropy loss, for both 10 and 100 step attacks. Further, we observe that we obtain stronger adversaries while optimising the margin loss between predicted probability scores, as compared to the corresponding logits. The weighting factor for the squared $\ell_2$ relaxation term is linearly decreased to 0 for the 100-step attack, while it is kept constant for the 10-step attack. From the 100-step evaluations, we observe that graduated optimisation indeed aids in finding stronger adversaries. Further, the addition of initial Bernoulli random noise aids in improving 100-step adversaries. We also note that GAMA-FW achieves the strongest attack when the available budget on the number of steps for attack is relatively small, making it suitable for use in multi-step adversarial training.

## 5.2 Variation of Accuracy and Cross-Entropy loss across attack iterations

In Fig.3, we plot the accuracy and cross-entropy loss across attack iterations for the TRADES ResNet-18 model on the CIFAR-10 dataset. The GAMA attack achieves lower accuracy and higher cross-entropy loss during the course of optimization, as compared to the attack generated using only the maximum-margin loss. We note from Fig.3(b) that the decay of $\ell_2$ relaxation term over the first 25 iterations in GAMA is crucial to allow cross-entropy loss to increase. Therefore, while the $\ell_2$ relaxation term gives the right initialization, switching to the true maximum-margin optimization objective is important for achieving a stronger attack. We thus observe that the additional $\ell_2$ relaxation term with a decaying coefficient indeed aids in the optimization process, and prevents the attack from stalling at points where the primary objective function attains a local maximum. Lastly, although the loss tends to oscillate before the first drop in step-size at iteration 60, we find that it is important to allow the attack to adequately explore the constraint set, in order to identify strong adversarial perturbations towards the end of optimization.

## 5.3 Use of ADAM optimizer in the GAMA attack

We also implement an ablation of the proposed GAMA-PGD attack using the ADAM optimizer [8] with the true gradients, instead of using Stochastic Gradient Descent with signed gradients. We perform a hyperparameter search over the initial step-size, step-schedule and decaying coefficient of the $\ell_2$ smoothing term. We find that the attack is marginally weaker when the ADAM optimizer is used; the strongest 100-step attack obtained over the entire hyperparameter search achieves 53.66% accuracy on the TRADES WideResNet-34 model for a single run of the attack, compared to 53.29% as obtained by the original GAMA-PGD attack.

## 6 Details on Evaluation of the proposed defense

In this section, we present additional experimental results to support our claims on the proposed single-step defense GAT. For CIFAR-10 dataset, we report results on ResNet-18 and WideResNet-34-10 architectures for the proposed method in the main paper. We note that while defense methods such as CURE [12] and 2-step LLR [13] achieve non-trivial robustness against multi-step adversaries using

Figure 3: **Variation of Accuracy and Cross-Entropy Loss across attack iterations:** Plot of the Accuracy and Cross-Entropy Loss across attack iterations for the TRADES ResNet-18 defense, trained on the CIFAR-10 dataset. The proposed attack GAMA (Maximum-margin + $\ell_2$ relaxation term) has been compared with a standard maximum-margin based attack.

Table 3: **GAT (Defense) Ablations (CIFAR-10)**: Accuracy (%) of various ablations of the GAT defense (rows) trained on ResNet-18 model under the $\ell_\infty$ threat model of $\varepsilon = 8/255$.

| Ablations | Clean | PGD-100 | AA |
|---|---|---|---|
| GAT (Proposed method) | 80.49 | 53.08 | 47.30 |
| **A1:** GAT, without alternating between CE and GA-CE attacks | 80.22 | 51.50 | 46.52 |
| **A2:** GA max-margin loss for attack and training | 23.22 | 15.64 | 10.98 |
| **A3:** GA-CE attack + standard defense (training on $CE_{clean} + CE_{adv}$) | 90.21 | 33.57 | 32.29 |
| **A4:** Standard attack + standard defense (R-FGSM training) | 89.24 | 34.23 | 33.16 |
| **A5:** Standard (CE) attack + GAT defense | 80.05 | 51.80 | 44.21 |

adversarial training on two or three step attacks, they are significantly weaker than recent single-step defense methods such as FBF [19] and R-MGM [18]. Thus, we restrict our primary comparisons to the latter defenses which are more robust for a similar computational budget. We use the GAT defense trained on the ResNet-18 architecture for further evaluations in this section.

## 6.1 Ablation Experiments

We present ablations on the proposed defense GAT, trained on CIFAR-10 dataset in Table-3. The architecture of the models is ResNet-18, and the models are trained to be robust under an $\ell_\infty$ threat model of $8/255$. We present results against PGD-100 step attack and the recently proposed ensemble of attacks, AutoAttack [4]. In order to diversify the attacks generated for GAT training, we switch between standard cross-entropy loss and the proposed GA-CE loss (Algo.-1, L7) in alternate iterations. However, even without this additional diversification step (Ablation-**A1**), we observe similar accuracy on AA with merely a marginal drop. We observe that using the GAMA loss (Eq.1) for both generation of attack and adversarial training does not lead to improved robustness (Ablation-**A2**). This is because minimization of maximum-margin objective is not suitable for training Deep Neural Networks.

The proposed defense involves the use of a modified loss function for both attack generation and adversarial training. We perform experiments to evaluate the impact of each of these components individually. The model in Ablation-**A3** is trained using GA-CE attack based adversaries. Adversarial training in this experiment in done by minimizing cross-entropy loss on both clean and adversarial samples, in similar vein to R-FGSM adversarial training. While R-FGSM training (Ablation-**A4**) leads to an improvement over this method, it is still significantly weaker than the proposed defense. Similarly, we find that using the GAT loss for defense alone (Ablation-**A5**) does not lead to significantly improved robustness. Therefore a combined usage of the loss in both attack generation and adversarial training is crucial for the state-of-the-art results obtained using GAT.

Figure 4: Plot of the loss surface of different models on perturbed images of the form $x^* = x + \delta_1 g + \delta_2 g^{\perp}$, obtained by varying $\delta_1$ and $\delta_2$. Here $g$ is the sign of the gradient direction of the loss with respect to the clean image ($x$) and $g^{\perp}$ is a direction orthogonal to $g$. The cross-entropy (CE) loss surface is plotted in the first row, and GAMA loss surface is plotted in the second row.

## 6.2 Stability of Guided Adversarial Training

In this section, we investigate the stability of the proposed training algorithm on the CIFAR-10 dataset. We train a ResNet-18 model multiple times allowing different random initialisation of network parameters in each run. For each run, we follow the training methodology as outlined in Sections 3.2 and 3.3. We observe that models trained using GAT are very stable; the PGD-100 accuracy obtained over six random reruns are as follows: 53.08, 53.21, 53.26, 52.46, 53.14, 53.03. The low variance (Standard Deviation = 0.266) across multiple runs highlights the stability of the proposed training method. Further, we note that models trained using GAT do not suffer from catastrophic overfitting, as observed in prior works such as FBF [19]. In Fig.2, we observe that the cross-entropy loss on adversaries generated using an FGSM attack is highly similar to the loss on PGD 7-step adversaries throughout the entire training regime, indicating the absence of catastrophic overfitting. We also observe that even the model obtained in the last epoch of training achieves high adversarial accuracy against strong multi-step attacks, in sharp contrast to models obtained towards the end of training using FBF.

## 6.3 Loss Surface Plots

To verify the absence of gradient masking, we visualise the loss surface of models trained using the proposed single-step defense GAT, in the neighbourhood of a test data sample. To generate the loss surface, we plot the loss obtained by perturbing the clean sample $x$ along two directions: one along the direction of the gradient of the loss at sample $x$, and another direction that is orthogonal to the gradient. In Fig.4 (a), (b) and (c), we plot the standard cross-entropy loss for FGSM-AT, PGD-AT and GAT trained models respectively. We find that FGSM training produces models with significant gradient masking and a convoluted loss surface. On the other hand, for PGD-AT and GAT models, the loss surface is smooth, thereby verifying the absence of gradient masking in the proposed single-step defense. In the second row of Fig.4, we plot the proposed GAMA loss (Eq.1 of the main paper), which is a combination of the maximum margin loss and the squared $\ell_2$ distance between the softmax predictions of the perturbed image and original data sample respectively. The squared $\ell_2$ relaxation term is weighted by a constant value of $\lambda = 25$ for obtaining the loss surface plots in Fig.4 (d), (e) and (f). We again find that the loss surface for the proposed defense GAT is smooth, despite being a single-step defense method. We further note that the GAMA loss surface is smoother than the cross-entropy loss surface for all three models, specifically for the FGSM model. This helps

Table 4: **Defenses (ImageNet-100)**: Accuracy (%) of different defenses (rows) trained on ResNet-18 architecture against various white-box attacks (columns) constrained within an $\ell_\infty$ radius of $8/255$. The first partition corresponds to single-step defenses, and the second to multi-step defenses. For the Carlini and Wagner (C&W) attack, the Mean-$\ell_2$ norm required to achieve high Fooling Rate (FR) is reported. Higher the $\ell_2$ norm, better is the robustness.

| Method | Clean Acc (%) | FGSM | IFGSM 7-step | PGD (n-steps) 7 | 20 | 500 | GAMA PGD-100 | AA | C & W Mean $\ell_2$ |
|---|---|---|---|---|---|---|---|---|---|
| Normal | **81.44** | 8.22 | 0.08 | 0.06 | 0.02 | 0.00 | 0.00 | 0.00 | 0.570 |
| RFGSM-AT [15] | 78.46 | 32.04 | 23.52 | 21.64 | 15.86 | 13.88 | 13.38 | 12.96 | **2.960** |
| FBF [19] | 57.32 | 36.24 | 26.92 | 29.84 | 28.00 | 27.22 | 21.78 | 20.66 | 0.737 |
| R-MGM [18] | 64.84 | 40.80 | 35.18 | 35.60 | 32.48 | 31.68 | 27.46 | 27.68 | 1.636 |
| GAT (**Ours**) | 67.98 | **45.38** | **39.66** | **40.18** | **38.02** | **37.46** | **29.30** | **28.92** | 1.499 |
| PGD-AT [11] | 68.62 | 43.04 | 40.00 | 39.64 | 37.20 | 36.56 | 32.24 | 32.98 | 1.550 |
| TRADES [21] | 62.88 | 40.46 | 38.52 | 38.44 | 37.34 | 37.24 | 31.44 | 31.66 | 1.360 |

Table 5: **Defenses (MNIST)**: Accuracy (%) of different defenses (rows) trained on M-LeNet architecture (Table-1) against various white-box attacks (columns) constrained within an $\ell_\infty$ radius of $0.3$. The first partition corresponds to single-step defenses, and the second to multi-step defenses. For the Carlini and Wagner (C&W) attack, the Mean-$\ell_2$ norm required to achieve high Fooling Rate (FR) is reported. Higher the $\ell_2$ norm, better is the robustness.

| Method | Clean Acc (%) | FGSM | IFGSM 40-step | PGD (n-steps) 40 | 100 | 500 | GAMA PGD-100 | AA | C & W Mean $\ell_2$ |
|---|---|---|---|---|---|---|---|---|---|
| Normal | 99.20 | 16.59 | 0.48 | 0.02 | 0.00 | 0.00 | 0.00 | 0.00 | 1.42 |
| RFGSM-AT [15] | **99.37** | 92.44 | 89.47 | 90.24 | 85.85 | 85.32 | 83.64 | 82.28 | 2.19 |
| FBF [19] | 99.30 | **97.47** | 94.53 | 94.85 | 92.35 | 91.37 | 87.27 | 79.02 | 1.91 |
| R-MGM [18] | 99.04 | 96.35 | 93.09 | 93.06 | 90.96 | 90.56 | 88.13 | 86.21 | **2.31** |
| GAT (**Ours**) | **99.37** | 97.11 | **95.61** | **96.11** | **94.58** | **94.44** | **92.96** | **90.62** | 2.30 |
| PGD-AT [11] | 99.27 | 96.27 | 94.91 | 95.53 | 94.14 | 93.98 | 92.80 | 91.81 | 2.63 |
| TRADES [21] | 99.32 | 96.08 | 94.86 | 95.26 | 93.52 | 93.40 | 92.74 | 92.19 | 2.53 |

explain why the proposed GAMA attack is more effective than the standard maximisation of the cross-entropy loss.

## 6.4 Performance against White-Box attacks

The results on white-box adversarial attacks for ImageNet-100 and MNIST datasets are presented in Table-4 and Table-5 respectively. On both datasets, we observe significant improvement in robustness with the proposed approach when compared to existing single-step adversarial training methods. We also note that the robustness achieved is comparable to the multi-step adversarial training methods, TRADES and PGD-AT, presented in the second partition of both tables.

We also evaluate all the defenses on MNIST and ImageNet-100 datasets against the proposed GAMA-PGD attack. The proposed defense is stronger compared to all other defenses even on the GAMA-PGD 100-step attack. The proposed GAMA-PGD attack is notably stronger than all the single attacks considered here. We note that for ImageNet-100 dataset, a single run of the GAMA-PGD attack is comparable to the AA attack, which is an ensemble of multiple attacks with five random restarts each. In particular, the GAMA-PGD attack is significantly stronger than the APGD-CE, APGD-DLR, FAB and Square attacks that constitute the AA ensemble attack. For MNIST dataset, GAMA-PGD is comparable to AA on some defenses and significantly weaker than AA on few others. This is primarily because one of the attacks in the AA ensemble is the Square attack, which is a query based attack. This is a gradient-free attack and is therefore significantly stronger than gradient-based attacks in cases where the loss surface is complex, leading to masking of the true gradient direction.

While the defense is trained to be robust against $\ell_\infty$ norm bound perturbations, we find that the robustness to the $\ell_2$ norm based Carlini & Wagner (C&W) attack [2] is comparable to other $\ell_\infty$ norm based adversarial training methods. We also evaluate the proposed method (GAT with ResNet-18

Table 6: Prediction accuracy (%) of our model (GAT) in various targeted and untargeted White-Box attack settings. Among the targeted attacks, we consider 1000-step Least Likely attack and 1000-step Random target attack. In the second partition, worst case robustness against multiple random restarts is reported. We consider a 1000-sample subset of CIFAR-10 and ImageNet-100 datasets for the random restarts experiments.

| Attack | CIFAR-10 | | ImageNet-100 | | MNIST | |
|---|---|---|---|---|---|---|
| | 500-step | 1000-step | 500-step | 1000-step | 500-step | 1000-step |
| PGD-Targeted (Least Likely class) | 79.50 | 79.50 | 66.12 | 66.02 | 99.03 | 99.03 |
| PGD-Targeted (Random class) | 74.56 | 74.37 | 63.80 | 64.10 | 98.86 | 98.84 |
| PGD-Untargeted | 53.04 | 53.04 | 37.52 | 37.52 | 94.37 | 94.37 |
| | 1-RR | 1000-RR | 1-RR | 500-RR | 1-RR | 1000-RR |
| PGD 50-step, r-RR | 53.20 | 52.10 | 38.70 | 38.30 | 95.46 | 92.20 |

Table 7: Prediction accuracy (%) of different defenses in **FGSM Black-Box attack** setting. Source model for attack is specified in the column headings. Target model is specified in each row.

| Method | CIFAR-10 | | | ImageNet-100 | | | MNIST | | |
|---|---|---|---|---|---|---|---|---|---|
| | Clean | VGG11 | ResNet18 | Clean | AlexNet | ResNet18 | Clean | BB-MNIST | M-LeNet |
| Normal | 92.30 | 37.09 | 15.98 | 81.44 | 63.82 | 8.22 | 99.05 | 38.62 | 16.58 |
| RFGSM-AT [15] | 89.66 | 82.62 | 85.74 | 78.46 | 74.82 | 74.18 | 99.37 | 93.79 | 92.44 |
| FBF [19] | 82.83 | 78.19 | 80.41 | 57.32 | 56.12 | 56.22 | 99.30 | 95.56 | 95.19 |
| R-MGM [18] | 82.29 | 78.18 | 79.99 | 64.84 | 63.26 | 63.60 | 99.04 | 95.52 | 95.19 |
| GAT (**Ours**) | 80.49 | 76.95 | 78.54 | 67.98 | 65.94 | 65.98 | 99.37 | 96.51 | 96.49 |
| PGD-AT [11] | 82.67 | 78.91 | 80.53 | 68.62 | 67.02 | 67.34 | 99.27 | 95.68 | 96.27 |
| TRADES [21] | 81.73 | 78.28 | 79.65 | 62.88 | 61.42 | 61.42 | 99.32 | 96.49 | 96.08 |

architecture) on the DDN attack [14] for CIFAR-10 dataset, and obtain a mean $\ell_2$ norm of 0.805 for adversarial perturbations, compared to 0.762 as obtained with the Carlini and Wagner (C&W) $\ell_2$ attack, indicating that the latter is stronger. Thus, we primarily utilise the C&W attack for the evaluation of defense models on $\ell_2$ norm-constrained adversaries.

We evaluate our proposed approach on 1000-step PGD targeted and untargeted attacks. The results of these experiments are presented in Table-6. Targeted attacks are weaker than untargeted attacks, thereby resulting in a higher accuracy. We note that the attack converges within 1000-steps based on the observation that drop in accuracy between 500-step attack and 1000-step attack is marginal.

We present the worst-case accuracy of GAT-trained models across multiple random restarts of PGD 50-step attack in the second partition of Table-6. The results are shown on a 1000-image subset of CIFAR-10 and ImageNet-100 test sets and on the full MNIST test set. In order to find the worst-case accuracy, we continue restarts until accuracy stabilizes. We note that the robustness of the proposed defense is not broken by attacks using random restarts, thereby demonstrating the absence of gradient masking.

## 6.5 Performance against Black-Box and gradient-free attacks

The results on FGSM and PGD 7-step Black-box attacks are presented in Table-7 and Table-8 respectively. We consider two sources for Black-box attacks; the first with the same architecture as the source model, and the second with a different architecture. Across all three datasets, accuracy on black-box attacks closely tracks the accuracy on clean samples, and is higher than that of white-box attacks. This confirms that there is no issue of gradient masking in the proposed model, which could potentially generate weaker adversaries, thereby creating a false sense of robustness.

We present results on the query-based black-box attack, Square [1] in Table-9. We note that across most defenses, this attack is weaker than PGD-500 step attack for CIFAR-10 and ImageNet-100, whereas it is stronger than the same for MNIST. The proposed defense achieves improved robustness compared to existing single-step adversarial defenses on MNIST and CIFAR-10 datasets. For ImageNet-100, although R-FGSM achieves better accuracy, it is significantly more susceptible to

Table 8: Prediction accuracy (%) of different models against **PGD Black-Box attacks**. Source model for attack is specified in the column headings. Target model is specified in each row. 7-step attacks are used for CIFAR-10 and ImageNet-100, and 40-step attacks are used for MNIST.

| Method | CIFAR-10 | | | ImageNet-100 | | | MNIST | | |
|---|---|---|---|---|---|---|---|---|---|
| | Clean | VGG11 | ResNet18 | Clean | AlexNet | ResNet18 | Clean | BB-MNIST | M-LeNet |
| Normal | 92.30 | 20.88 | 0.00 | 81.44 | 72.48 | 0.04 | 99.05 | 8.03 | 0.01 |
| RFGSM-AT [15] | 89.66 | 84.96 | 87.36 | 78.46 | 76.58 | 76.24 | 99.37 | 94.93 | 89.47 |
| FBF [19] | 82.83 | 79.62 | 81.32 | 57.32 | 56.66 | 56.84 | 99.30 | 96.53 | 96.51 |
| R-MGM [18] | 82.29 | 79.33 | 80.92 | 64.84 | 63.88 | 64.20 | 99.04 | 96.03 | 96.28 |
| GAT (**Ours**) | 80.49 | 77.88 | 79.25 | 67.98 | 66.58 | 66.80 | 99.37 | 97.25 | 97.52 |
| PGD-AT [11] | 82.67 | 79.68 | 81.26 | 68.62 | 67.66 | 67.90 | 99.27 | 96.46 | 94.91 |
| TRADES [21] | 81.73 | 79.04 | 80.22 | 62.88 | 62.28 | 62.26 | 99.32 | 96.98 | 94.86 |

Table 9: Prediction accuracy (%) of different defenses against the **Square Attack**, which is a query-based Black-Box attack.

| Method | CIFAR-10 | | ImageNet-100 | | MNIST | |
|---|---|---|---|---|---|---|
| | Clean | Square | Clean | Square | Clean | Square |
| Normal | 92.30 | 0.16 | 81.44 | 0.66 | 99.05 | 0.02 |
| RFGSM-AT [15] | 89.66 | 43.01 | 78.44 | 41.60 | 99.37 | 84.13 |
| FBF [19] | 82.83 | 52.46 | 57.32 | 28.38 | 99.30 | 79.66 |
| R-MGM [18] | 82.29 | 52.87 | 64.82 | 38.34 | 99.04 | 88.89 |
| GAT (**Ours**) | 80.49 | 53.62 | 67.98 | 39.06 | 99.37 | 91.03 |
| PGD-AT [11] | 82.67 | 52.64 | 68.60 | 45.50 | 99.27 | 91.99 |
| TRADES [21] | 81.73 | 54.87 | 62.86 | 40.30 | 99.32 | 92.60 |

white-box attacks presented in Table-4. The performance of the proposed single-step defense GAT against the strong query based Square attack shows that the robustness of the proposed defense is not a result of gradient masking.

Further, we evaluate the proposed defense against SPSA [17], a gradient-free attack that utilises a numerical approximation of gradients by sampling function values along random directions. We use the following set of hyperparameters for SPSA attack generation: learning rate $= 0.01$, $\delta = 0.01$, number of iterations $= 100$ and number of samples to approximate the average gradient $= 128$. On the CIFAR-10 dataset, the proposed single-step GAT defense with ResNet-18 architecture achieves $56.59\%$ accuracy against SPSA, compared to $53.62\%$ on the Square attack. Since the Square attack is stronger than SPSA, we use the former to present gradient-free attack evaluation of all defense methods in Table-9.

Table 10: Prediction accuracy (%) of our proposed single-step defense GAT against **Adaptive Attacks** constructed using diverse settings (rows).

| $\lambda_{attack}$ | Decay Iterations | CIFAR-10 | | ImageNet-100 | | MNIST | |
|---|---|---|---|---|---|---|---|
| | | GA-CE | GAMA | GA-CE | GAMA | GA-CE | GAMA |
| 0 | no decay | 52.93 | 48.40 | 36.66 | 30.14 | 93.22 | 93.06 |
| $\lambda_{defense,init}$ | no decay | 52.83 | 49.78 | 35.76 | 32.80 | 93.07 | 93.20 |
| $\lambda_{defense,final}$ | no decay | 53.63 | 51.74 | 35.64 | 35.18 | 93.09 | 93.15 |
| 50 | 25 | 52.80 | **47.76** | 36.58 | **29.24** | **93.02** | **93.00** |
| 100 | 25 | 52.75 | 47.92 | 36.66 | 29.34 | 93.13 | 93.05 |
| 50 | 50 | **52.73** | 47.88 | 36.56 | 29.40 | 93.20 | 93.02 |
| 100 | 50 | 52.84 | 47.97 | 36.58 | 29.30 | 93.23 | 93.11 |
| 50 | 100 | 53.00 | 50.64 | 36.58 | 32.22 | 93.19 | 93.15 |
| 100 | 100 | 53.34 | 51.68 | **35.52** | 33.54 | 93.39 | 93.35 |

Figure 5: Plot showing the variation of accuracy on n-step (n=7 for CIFAR-10 and ImageNet-100, n=40 for MNIST) PGD adversarial attacks across the magnitude of perturbation, $\varepsilon$ for the proposed single-step defense, GAT. Accuracy goes to zero for large $\varepsilon$ across all datasets, indicating the absence of gradient masking.

Figure 6: Plot showing the variation of loss on FGSM samples across the magnitude of perturbation, $\varepsilon$ for the proposed single-step defense, GAT. Loss increases monotonically with an increase in $\varepsilon$ across all datasets, indicating the absence of gradient masking.

## 6.6 Performance Against Adaptive Adversaries

Since we consider the framework of worst-case adversarial robustness, we assume that the adversary has complete knowledge of the defense mechanism employed. Thus, it is crucial to evaluate our model on adaptive adversaries as well [3]. We consider strong multi-step PGD attacks, where the adversary attempts to maximise the proposed loss at each step. More precisely, the adversaries are generated using the proposed GA-CE or GAMA loss, where the first term is either cross-entropy loss or maximum margin loss, and the second term is the squared $\ell_2$ distance between the softmax outputs of the original clean image and the adversarial sample generated in the previous step. We evaluate the proposed defense against diverse settings of the attack, using the cross-entropy loss in one case (GA-CE) and maximum-margin loss in the second (GAMA). We present our experiments and results in Table-10. We consider a case without $\lambda$ decay over iterations, where the $\lambda$ used in the attack ($\lambda_{attack}$) is same as the $\lambda$ used in defense ($\lambda_{defense,init}$). We also consider a case where $\lambda_{attack} = \lambda_{defense,final}$ which is the same as $\lambda_{defense,init}$ multiplied by the step-up factor. In general, the cross-entropy based attack is weaker than the margin based attack. The adaptive attacks are significantly stronger than standard PGD-based attacks. However, they are not as strong as the AutoAttack (AA). We note that AutoAttack is expected to be stronger since it is an ensemble of attacks, which declares a given data sample to be robust only if it passes all the attacks in the ensemble. Therefore, although we generate strong adaptive attacks, robustness of the proposed approach does not deteriorate further compared to AutoAttack.

## 6.7 Sanity checks to ensure absence of gradient masking

We observe from the above experiments that iterative attacks are stronger than single-step attacks (Table-2 in the main paper, Tables-4 and 5). Furthermore, white-box attacks are stronger than black-box attacks (Table-2 in the main paper, Tables-4, 5, 7 and 8). In the plot shown in Fig.5, we increase the value of $\varepsilon$ and observe that unbounded attacks are able to reach 100% attack success rate for all datasets. Also, as shown in Fig.6, the loss monotonically increases with an increase in the perturbation size of the FGSM attack. These tests confirm that the model is truly robust, and the observed robustness is not a result of gradient masking.