[Reviews · NeurIPS 2020]

Review 1

Summary and Contributions: The paper proposes to add a relaxation term to the losses (cross-entropy, margin loss) used for standard adversarial attacks to improve their effectiveness. Moreover, such attacks can be used at training time to produce, via adversarial training, robust models, especially with single-step methods.

Strengths: - The suggested additional term is simple and shown effective in practice. - The new loss allows to train with single-step first order attacks classifiers which are more robust than what achieved in previous works. - The evaluation of the proposed defenses includes many attacks and seems properly conducted. - The proposed attack, GAMA, is effective in evaluating the robustness of many defenses, outperforming existing individual attacks.

Weaknesses: - The paper proposes many different variations of the methods (different losses, training schemes, optimization schemes) which are used for different tasks, each with different parameters. Thus, it is sometimes difficult to follow which exact setup is used for every task. This also makes the method less general. - Some of the variations introduced seem a bit hacky, e.g. changing the loss for adversarial training in alternate iterations.

Correctness: The claims and method seem correct, and the experiments properly done.

Clarity: The paper is well written, although in some parts, especially in the appendix, some details could be slightly more clearly presented.

Relation to Prior Work: The paper introduces properly prior works.

Reproducibility: Yes

Additional Feedback: - In Eq. (1), shouldn't the margin loss have the opposite sign? In the current formulation, maximizing L as in Eq. (1) would increase the logit of the true class compared to the others (or am I missing something?). - In my point of view, the main contribution of the paper consists in training robust models with 1-step methods achieving better robustness than Wong et al. [31] (as acknowledged by the authors for multi-step adversarial training the improvement is minimal, and in my opinion not significant), and this could be further emphasized. In [31] the models also suffer from catastrophic overfitting, i.e. the robust training fails depending on different random seeds. How stable is 1-step GAT wrt different seeds? Does it always yields similar results or is there high variance? - On the line of my previous point, I think it'd be helpful and interesting to analyse in more details why using the proposed regularization helps 1-step adversarial training (my guess is that the loss landscape is made smoother so that FGSM is almost as effective as PGD). - According to what mentioned in the Appendix, for GAMA-PGD an initial step size of 2\epsilon is used and later reduced. This seems similar to what proposed in APGD. Have you maybe tried to use the proposed loss within the APGD framework? Overall, I think the paper proposes an effective method. In my opinion, the presentation could be improved, since, as said above, the many variations give the impression that many methods are proposed for different tasks. ### Update post rebuttal ### I thank the authors for the detailed response. My opinion about the paper remains unchanged. I invite the authors to improve the clarity of the presentation and the discussion of prior works (in particular, I think a comment about the loss function of [29] could be useful, as it shares some similarities with the proposed one) for the benefit of the reader.


Review 2

Summary and Contributions: The authors propose adversarial attacks that incorporate an additional relaxation term to the standard loss as used in existing attacks. The attacks can be used to generate adversarial examples to train robust models. Experimental results show that the proposed method outperforms other single-step training methods.

Strengths: + This paper rethinks the optimization procedure of PGD and proposes a new method that outperforms PGD in certain scenarios. + Extensive experiments have been performed for testing the proposed method.

Weaknesses: I'm confused about the objective function of the proposed attacks in Eqn (1). Let us consider generating an adversarial example using a clean image for initialization. In the first step, the third term in Eqn (1) is zero, so the first two terms dominate the loss and the first optimization step minimizes the confidence score of the label class and maximizes the confidence score of one other class. Then in the second step, the third term in Eqn (1) plays a similar role to that of the maximum margin loss. Table 2 in the paper also demonstrates that the proposed l2 loss help to a very limited extent and it seems that the maximum margin loss developed in CW's attack [1] contributes much more. Can the authors provide more discussions about how the proposed l2 loss helps theoretically and empirically? I have checked the submitted code and found that the l2 loss is seemingly encouraged to be minimized in the code implementation, rather than being maximized as described in the paper. [1] N. Carlini and D. Wagner. Towards evaluating the robustness of neural networks. In IEEE Symposium on Security and Privacy (SP). IEEE, 2017. -------------------Post rebuttal-------------------------- I would like to thank the authors for response to my comments. My concerns have been partially addressed and thus I'm happy to raise the score to accept.

Correctness: See the box above.

Clarity: Yes, the paper is well written.

Relation to Prior Work: The connection betweent the proposed method and CW's attack should be further discussed.

Reproducibility: No

Additional Feedback:


Review 3

Summary and Contributions: This manuscript introduces a modified loss function for PGD adversarial attacks that finds more suitable gradient-directions, increases attack effectiveness and leads to more effective adversarial training.

Strengths: The method is extensively compared to several other SOTA attacks and adversarial training methods on a battery of different neural network models. The results are promising, especially taking into account the much lower computational overhead of the method compared to standard PGD. I am not aware of a similar loss function proposed in the literature.

Weaknesses: It would be good to get a better theoretical understanding of the regularization term and its effect on the optimisation. Under what conditions can you expect that the modified gradients are more closely aligned with the "optimal" descent direction? Why is this regularization term better than e.g. the cross-entropy term (which also takes into account all latents and should be more biased towards latents with more sensitive probability scores)?

Correctness: The results look sensible and the reported values for competing methods match the literature, except maybe FAB on Madry et al for which [1] reports 45.37% instead of 50.67% as reported here. Maybe the authors can comment on this deviation. [1] Reliable evaluation of adversarial robustness with an ensemble of diverse parameter-free attacks, https://arxiv.org/abs/2003.01690

Clarity: The paper is overall well written and the story line is easy to follow.

Relation to Prior Work: Overall the manuscript reviews most related work, although it could include a bit more the wider range of methods SPSA [1], L2 attacks like DDN [2] or the Brendel & Bethge attack [3]. The latter two also tend to be quite a bit stronger than C&W in practice, which could strengthen the L2 evaluation in table 3. [1] https://arxiv.org/pdf/1802.05666.pdf [2] https://arxiv.org/pdf/1811.09600.pdf [3] https://papers.nips.cc/paper/9446-accurate-reliable-and-fast-robustness-evaluation.pdf

Reproducibility: Yes

Additional Feedback: * What is the initial value for lambda? How sensitive is the method to different choices. --Post-Rebuttal Statement The rebuttal addressed many concerns raised by me and others, and I am happy to keep my positive assessment of the work.


Review 4

Summary and Contributions: *********************Update*********************** I want to thank the authors for their response. Many of which sufficiently addresses my concerns. I am changing my score to a 6. The reason why I am not changing this to a 7 is for an empirical-based paper that is less theoretical I would have expected slightly more analysis (not just on single example landscapes or ablation studies), but more detailed analysis/careful design of experiments to shed light on what the regulariser might actually be doing to the neural network. Either way, I think the regulariser is something easy to try, the results for one-step is very good. Thus I will change this to a 6. ****************************************************************** This paper introduces a regularisation term on to the objectives used for adversarial attacks and demonstrate its efficacy when it is used for both defence and attack scenarios. Crucially, they show that this is an objective which can be used for a single-step attack defence which previously has been demonstrated to have effects of gradient obfuscation.

Strengths: Empirically, it seems to achieve good adversarial accuracy with a single step of PGD, namely 49% adversarial accuracy on CIFAR-10. This seems like a scalable method for tasks such as ImageNet.

Weaknesses: I'm a little concerned about the strength of the adversarial attack introduced, the key reason for my worry is highlighted in the correctness section below. For the one-step defense, the nominal accuracy (80%) for CIFAR-10 seems a bit too low for WRN-34-10. I thought this should be above 83% at the very least. Does the authors have an intuition as to why the nominal accuracy is so low? One axes which this has missed is the optimiser used for the objective. How does this adversarial attack compare to the margin loss with Adam optimisation? Or even using the gradient rather than the sign of the gradient. There are no ablations on how the amount regularisation term added onto the loss affect the attack attack and very little theoretical/empirical analysis on how this regularisation term will guide the attack. In other words, there is no theoretical grounding as to why this would be a better objective for optimisation, it seems like an empirical argument - which is also fine but then more analysis would be needed to back up the intuition. I see that Figure 1 gives an example of when this regularisation is useful, but does this example hold true in practise?

Correctness: I guess what was very striking in terms of the results section is the adversarial accuracy they obtain using their attack for Madry et al's network. For Multi-Targeted attacks and AutoAttack, the accuracy obtained is around 44%. But for their "strong" attack, they can only obtain 49.81 - 50%? This is worrying ... even in their Table 3 they show that the adversarial training obtains 44, why is the adversarial accuracy reported for their attack on Madry et al's network so high in Table 1? For Multi-Targeted attack in Figure 2, the paper shows that using 1 random restart we can only obtain 66% adversarial accuracy for CIFAR-10, 8/255. From my experience of multi-targeted it definitely gets below 60%, it should be around 55% or lower. It might be because the authors used only the sign of the gradient and wrong hyperparameters, note that in the multi-targeted paper many of the experiments was done with Adam optimisation. Another comment is that for Multi-Targeted attack comparison in Table 1, the choice was to choose 5 random targets whereas in the MT paper they have chosen the targets by using the second/third etc etc highest logits. This should be changed. I think Eq. 1 has some signs wrong? The objective is one which is maximised (as they have explained in the paper and also show this in the algorithm), but then the objective is maximising the probability of the true label while minimising the probability of other labels which is wrong. Unless the objective is in fact a targeted attack? In which case the text doesn't match the equation. Given this is the only equation in the entire paper, the authors should not have got this wrong. The loss surface plots definitely seems to show that GAT produces non-obfuscated loss surface but it was not specified whether these plots generated are for single-step GAT or multi-step GAT as the claim that their single-step methodology does not cause gradient obfuscation should be backed up with loss surface plots as well.

Clarity: The introduction and related section are well written. The tables are formatted in a way that it is hard to distill what exactly is the advantage of this method. Maybe some of these tables can be moved to the appendix and make a smaller version in the main text that makes the delta in performance much easier to read and understand.

Relation to Prior Work: Regarding single step methods I think this paper also needs to be compared methods that are also cheap computationally such as CURE [1] and LLR [2] both of which just needs 2/3 gradient calculations to avoid gradient obfuscation. [1] Moosavi-Dezfooli, Seyed-Mohsen, et al. "Robustness via curvature regularization, and vice versa." Proceedings of the IEEE Conference on Computer Vision and Pattern Recognition. 2019. [2] Qin, Chongli, et al. "Adversarial robustness through local linearization." Advances in Neural Information Processing Systems. 2019.

Reproducibility: Yes

Additional Feedback: Some of my comments were regarding baselines for the adversarial attack but it might be due to the fact the authors just used the sign of the gradient, I think it is crucial for the authors to revisit using optimisers such as Adam to test if their adversarial attack can be even stronger. It is also very important to make sure that the baselines are what we should expect, currently it doesn't seem to be the case. I hope the authors make sure this is fully addressed.

[Author Response · NeurIPS 2020]

We sincerely thank the reviewers for their time and valuable feedback on our work. We are pleased to see that the
reviewers find our work interesting, thorough and well-written. We thank **R1** and **R3** for their motivating comments on
the proposed single-step defense. We will emphasize this more in the final version. We sincerely apologize for the error
in sign of the max-margin term in the loss (Eq.1 in main paper). We understand that this has led to significant confusion.
The corrected loss which is maximized for attack generation is : $L = -f_\theta^y(\widetilde{x}) + \max_{j \neq y} f_\theta^j(\widetilde{x}) + \lambda \cdot ||\boldsymbol{f_\theta}(\widetilde{x}) - \boldsymbol{f_\theta}(x)||_2^2$

Discussion on the proposed regularizer: We justify the significance of proposed regularizer for GAT defense in Sec.1 of
the Suppl. This can be extended to attacks as well. The local Lipschitz constant ($\mathcal{L}$) of adversarially trained models is
low compared to standard models. Based on Eq.5 in the Suppl., $\mathcal{L}$ acts as an upper bound to the $\ell_2$ term upto a constant
factor. Hence, a low value of $\mathcal{L}$ leads to a low value of the $\ell_2$ term. Therefore, while finding an adversary, maximization
of the $\ell_2$ term additionally leads the optimization to move towards the direction of worst case local smoothness. The
use of $\ell_2$ term is also motivated by the use of a better optimization objective initially as discussed in L168-L180 of
main paper. We will explain these in more detail in the final version. The plot of CE loss vs. iterations (will be included
in final version) for the proposed attack shows a larger increase in CE loss in presence of the $\ell_2$ term. We will also draw
parallels with the theory of graduated optimization (On Graduated Optimization for Stochastic Non-Convex Problems,
Hazan et al.), which shows that such methods can lead to improved optimization for the family of $\sigma$-nice functions.

**[R1]** Too many variations of proposed method: We thank **R1** for the feedback. We will certainly work on improving
the clarity of experimental setup. Although we proposed multiple variants, we would like to clarify that the main attack,
GAMA-PGD uses the same loss function (max-margin, $\ell_2$ term) and optimizer (PGD) across all experiments. Also, the
main defense, GAT uses the same optimizer (single-step PGD) and loss (CE, $\ell_2$ reg) across all experiments.
**[R1]** Loss change in alternate iterations seems hacky: Results in Table-2 of the Suppl. show that impact of alternating
losses is marginal. The AA accuracy is $46.37\%$ without alternation and $46.72\%$ with alternation. (L169-172 of Suppl.)
**[R1]** Stability of GAT across reruns: We get similar results with low variance (SD = 0.224). The PGD100 CIFAR10 acc
across reruns are 52.14, 51.7, 52.02, 52.35, 51.96, 51.74. Unlike FBF, even in the last epoch, we obtain robust models.
**[R1]** Use of APGD framework: We thank **R1** for the valuable suggestion. We will certainly investigate this in future.

**[R2]** Objective function in Eq.1: We request **R2** to kindly reconsider the contributions of our paper after the correction
of loss function in L3-L5 above. The $\ell_2$ regularizer is maximized for attack generation and minimized in the defense.
**[R2]** Comparison to CW attack, significance of $\ell_2$ term in attack: CW attack uses max-margin loss in logit space, while
we use this in softmax space. We introduce the $\ell_2$ regularizer which is decayed to 0 over a few iterations. The advantage
of the proposed approach is not only the addition of $\ell_2$ loss term, but also in decaying it to 0 over a few iterations.
Therefore, from Table-2 in main paper, the difference (100-step, 1 run) w.r.t. CW attack is $0.9\%$ and advantage from the
$\ell_2$ regularizer and its schedule is $0.65\%$, both of which are significant relative to the trends on attack leaderboards. We
get a significant boost over CW attack across all defenses in Table-1. We will include these results in the final version.
**[R2]** Significance of $\ell_2$ term in defense: Table-2 in the Suppl. shows that without the $\ell_2$ term in adversary generation,
the AA accuracy is $43.37\%$, while it increases to $46.37\%$ with the $\ell_2$ term included. Similarly, by replacing the $\ell_2$ term
in defense with CE on adv samples, AA accuracy drops to $30.2\%$, which is $16.52\%$ lower than the proposed method.

**[R3]** SPSA, $\ell_2$-attacks: We thank **R3** for the valuable suggestions. We report results against the gradient-free attack,
Square in the paper. We will certainly include results on SPSA and the suggested $\ell_2$ attacks in the final version.
**[R4, R3]** Choice of $\lambda$ and sensitivity for the attack: Kindly refer to Section-3.2 of the Suppl. and Fig.1(a) of the Suppl.
**[R4, R3]** Results on CIFAR-10 defense by Madry et al.: We consider the ResNet-50 (not WRN-34) architecture for
reporting results on the defense by Madry et al. We use the pretrained model available in their *robustness* GitHub repo.
However, the numbers reported in FAB, MT and AA papers are on the WRN-34 model by Madry et al. We apologize
for missing the architecture details of defenses in Table-1. We will certainly include it in the final version. We use
ResNet-18 architecture for the PGD-AT model in Table-3 since the same architecture is used across all defenses.
**[R4]** MT baseline results: For the plot in Fig.2(a), we cycled through the other 9 classes of CIFAR-10 in a random order
and the $10^{th}$ restart was an untargeted max-margin attack. With $2^{nd}$ highest logit as the first target, the single restart acc
is $54.33\%$. There is no change in the 10-restart accuracy as expected. We use Adam (without sign of gradient) and
other hyperparameters as used by the MT authors. For the 5-restart results (4 random targets + 1 untargeted) reported in
Table-1, we see marginal improvement with use of highest logits. For the Trades defense, MT attack acc improves from
$53.57\%$ to $53.32\%$ with the use of highest logits. GAMA-PGD achieves $53.17\%$ and an MT version of GAMA attack
achieves $53.09\%$ for 5 restarts. We thank **R4** for this feedback. We will update the table and plot in the final version.
**[R4, R1]** Loss landscape: Fig.3(c) in Suppl. shows that the loss landscape of the single-step defense GAT is smooth.
**[R4]** Clean acc of GAT: We use 40k-10k train-val split for GAT (single-step) training, whereas for other defenses, full
50k train set was used. With 49k-1k split on CIFAR10 WRN-34, we get clean acc = $85.17\%$ and AA acc = $50.27\%$.
**[R4]** We thank **R4** for the suggestions. We will explore the use of Adam for GAMA attack, include the baselines CURE
($41.4\%$ PGD-20 acc on WRN28-10, CIFAR10) and LLR ($44.5\%$ MT acc on WRN28-8, CIFAR10) and organize the
tables better. The proposed method GAT is significantly better than these baselines under limited budget constraints.
*We look forward to more insightful discussions on our work at NeurIPS 2020.*

[Meta-Review · NeurIPS 2020]

This paper introduces a new attack to generate adversarial examples by adding an additional relaxation to the loss term. The reviewers complemented the evaluation in the paper and liked the proposed method. This is a strong paper.